

# Quantification of lignin oxidation products as vegetation biomarkers in speleothems and cave drip water

Inken Heidke[1], Denis Scholz[2], and Thorsten Hoffmann[1]

[1]Institute of Inorganic Chemistry and Analytical Chemistry, Johannes Gutenberg University of Mainz, Duesbergweg 10-14, 55128 Mainz, Germany
[2]Institute of Geosciences, Johannes Gutenberg University of Mainz, J.-J.-Becher-Weg 21, 55128 Mainz, Germany

**Correspondence:** Thorsten Hoffmann (t.hoffmann@uni-mainz.de)

**Abstract.** Here we present a sensitive method to analyse lignin oxidation products (LOPs) in speleothems and cave drip water to provide a new tool for paleo vegetation reconstruction. Speleothems are valuable climate archives. However, compared to other terrestrial climate archives, such as lake sediments, speleothems contain very little organic matter. Therefore, very few studies on organic biomarkers in speleothems are available. Our new sensitive method allows to use LOPs as vegetation

biomarkers in speleothems.

Our method consists of acid digestion of the speleothem sample followed by solid phase extraction (SPE) of the organic matter. The extracted polymeric lignin is degraded in a microwave assisted alkaline CuO oxidation step to yield monomeric LOPs. The LOPs are extracted via SPE and finally analysed via ultrahigh-performance liquid chromatography (UHPLC) coupled to electrospray ionisation (ESI) and high-resolution orbitrap mass spectrometry (HRMS). The method was applied to

stalagmite samples with a sample size of $3-5$ g and cave drip water samples with a sample size of $100-200$ mL from the Herbstlabyrinth-Advent-Cave in Germany. In addition, fresh plant samples, soil water and powdered lignin samples were analysed for comparison. The concentration of the sum of eight LOPs ($\Sigma 8$) was in the range of $20-84$ ng $\cdot$ g$^{-1}$ for the stalagmite samples and $230-440$ ng $\cdot$ L$^{-1}$ for the cave drip water samples. The limits of quantification for the individual LOPs ranged from $0.3-8.2$ ng per sample.

Our method represents a new and powerful analytical tool for paleo vegetation studies and has great potential to identify the pathways of lignin incorporation into speleothems.

## 1 Introduction

Here we present a sensitive method to analyse the lignin composition of organic traces contained in speleothems. This method offers new possibilities for paleo vegetation reconstruction since it combines the advantages of lignin analysis as a highly

specific vegetation biomarker with the benefits of speleothems as unique terrestrial climate archives. The major advantage of speleothems is that they can be dated very accurately (Richards and Dorale, 2003; Scholz and Hoffmann, 2008). Up to now, lignin analysis for paleo vegetation reconstruction was only applied to lake sediments and peat cores, which contain much larger amounts of organic matter than speleothems.




Speleothems are calcareous mineral deposits that form within caves in karstified carbonate rock. The most common types of speleothems are stalagmites, which are formed by water dripping on the ground of the cave, stalactites, which are their counterparts on the cave ceiling, and flowstones, which are formed by water films flowing on the cave walls and floor. Speleothems preserve information about climatic and hydrological conditions and the vegetation development above the cave and therefore

serve as paleoclimate archives (Fairchild and Baker, 2012; McDermott, 2004). Compared to other paleo climate archives, such as ice cores and marine or lacustrine sediments, speleothems have certain advantages. They can grow continuously for $10^3 - 10^5$ years, their growth layers are mechanically undisturbed and they do not show a loss of time resolution with increasing age (Gałuszka et al., 2017; Fairchild et al., 2006). They can be accurately dated up to $500\,000$ years back in time using the $^{230}$Th/U-method (Scholz and Hoffmann, 2008). Moreover, they occur on all continents except Antarctica and are thus not

limited to certain climatic regions.

Most studies of speleothems focus on the analysis of stable isotope ratios ($\delta^{13}$C, $\delta^{18}$O (McDermott, 2004) and inorganic trace elements (Fairchild and Treble, 2009). The organic content of speleothems has so far mostly been analysed as total organic carbon content or fluorescent organic matter (Quiers et al., 2015). However, in recent years, the interest in molecular organic proxies in climate archives has increased (Giorio et al., 2018; Blyth et al., 2008; Blyth and Watson, 2009; Blyth et al.,

2010, 2016). In speleothems, in particular lipid biomarkers, such as fatty acids reflecting changes in vegetational and microbial activities (Xie, 2003; Blyth et al., 2006; Bosle et al., 2014) and membrane lipids (glycerol dialkyl glycerol tetraethers, GDGTs) serving as paleo temperature proxies (Blyth and Schouten, 2013; Baker et al., 2016), have been studied.

Lignin occurs almost exclusively in terrestrial vascular plants and is one of the main constituents of wood and woody plants (Jex et al., 2014). It is a biopolymer that mainly consists of three monomers: sinapyl alcohol, coniferyl alcohol and p-coumaryl

alcohol. The proportion of these three monomers varies with the type of plant, such as gymnosperm or angiosperm and woody or non-woody material. Thus, by analysing the composition of lignin, it is possible to determine the source and type of plant material.

Lignin has been widely used as paleo vegetation proxy in lake sediment (Tareq et al., 2011) and peat cores (Tareq et al., 2004). In marine sediments (e.g., Zhang et al., 2013) and natural waters (Standley and Kaplan, 1998; Hernes and Benner, 2002),

lignin analysis has been used to determine the source of dissolved organic matter. Blyth and Watson (2009) have successfully detected lignin pyrolysis products in speleothems by applying a tetramethylammonium hydroxide (TMAH) thermochemolysis method, but there have been no quantitative studies of lignin in speleothems yet.

Lignin has to be degraded before the molecular composition of its phenolic components can be analysed. The most common method for degradation of lignin is the alkaline oxidation with cupric oxide (CuO), developed by Hedges and Parker

in 1976. This method releases a number of phenolic acids, aldehydes and ketones, which can be divided into four groups: The vanillyl group (V) consisting of vanillic acid, vanillin and acetovanillone, the syringyl group (S) consisting of syringic acid, syringaldehyde and acetosyringone, the cinnamyl group (C) consisting of trans-ferulic acid and p-coumaric acid, and the p-hydroxyl group (P) consisting of p-hydroxybenzoic acid, p-hydroxybenzaldehyde and p-hydroxyacetophenone. Hedges and Mann (1979) analysed fresh plant tissues and showed that the phenols of the syringyl group are only obtained from angiosperm,

but not from gymnosperm plant tissues. Likewise, the phenols of the cinnamyl group are only obtained from non-woody and not





from woody plant tissues, whereas the phenols of the vanillyl group are found in all kind of vascular plant tissues (angiosperm and gymnosperm, woody and non-woody). These results led to the introduction of the lignin oxidation product (LOP) parameters C/V and S/V, where C, for example, is defined as the sum of all lignin oxidation products of the C-group (Hedges and Mann, 1979). The phenols of the p-hydroxyl group can originate from gymnosperm and non-woody angiosperm plant tissues,

but are also oxidation products of protein rich organisms such as bacteria and plankton. Therefore, the P group is not used in the parameters to determine the source of lignin (Jex et al., 2014). The parameter $\Sigma 8$ gives the sum of the eight analytes of the C, S and V-group and is used to estimate the total amount of LOPs in a sample.

The oxidation with CuO has been optimised many times in the past. For example, Goñi and Montgomery (2000) developed a microwave digestion method. Other groups improved the sample clean-up by replacing the formerly used liquid-liquid ex-

traction (LLE) with solid phase extraction (SPE) (Kögel and Bochter, 1985; Kaiser and Benner, 2012). As the CuO oxidation method is broadly used, there are many data sets to compare with. This is certainly an advantage compared to the above mentioned TMAH thermochemolysis method, which is less often used and produces more complex methylated reaction product mixtures (Wysocki et al., 2008). For the detection of the LOPs, gas chromatography coupled to mass spectrometry (GC-MS) is often used, which requires a derivatisation step. Liquid chromatography is also used, either in combination with UV detection

or coupled to mass spectrometry.

The purpose of this study was to develop and validate a new selective and accurate method for the quantification of LOPs in both speleothem and cave drip water samples using liquid chromatography electrospray ionisation mass spectrometry (LC-ESI-MS). The stalagmite samples are first acid digested, and the acidic solution is then extracted by SPE. The eluent is then subjected to CuO oxidation in a microwave assisted digestion method. The oxidised sample solutions are again extracted by

SPE, and the LOPs are then separated and detected by ultrahigh-performance liquid chromatography coupled to electrospray ionisation high-resolution mass spectrometry (UHPLC-ESI-HRMS).

## 2   Experimental section

### 2.1   Chemicals and materials

Analytical standards of acetosyringone (97%), acetovanillone ($\geq$ 98%), para-coumaric acid ($\geq$ 98%), ethylvanillin (99%), fer-

ulic acid (99%), para-hydroxyacetophenone ($\geq$ 98%), para-hydroxybenzaldehyde ($\geq$ 97.5%), syringaldehyde (98%), syringic acid (> 95%) and cinnamic acid (97%) as well as copper(II) oxide (> 99%) and ammonium iron(II) sulfate (99%) were purchased from Sigma Aldrich. Analytical standards of para-hydroxybenzoic acid (99%) and vanillin (99%) were obtained from Acros Organics, an analytical standard of vanillic acid (98%) was obtained from Alfa Aesar. Sodium hydroxide (pellets, $\geq$ 99%) was purchased from Carl Roth, hydrochloric acid (HCl, suprapure, 30%) from Merck KGaA. Lignin from mainly coniferous

wood was obtained from BASF SE. Mixed lignin from wheat straw and various kinds of wood was purchased from Bonding Chemical. Solid phase extraction columns (Oasis HLB, 3 mL tubes, 60 mg packing material) were purchased from Waters. Ultrapure solvents (Optima LC/MS grade) acetonitrile (ACN), water ($H_2O$) and methanol (MeOH) were obtained from Fisher



Scientific. Dichloromethane (DCM) ($\geq$ 99.9% (GC)) was obtained from Honeywell Riedel-de Haën. Ultrapure water with 18.2 M$\Omega$ resistance was produced using a Milli-Q water system from Merck Millipore (Darmstadt, Germany).

## 2.2 Methods

### 2.2.1 Preparation of standards

Stock solutions of all analytical standards were prepared at a concentration of 1 mg·mL$^{-1}$ in ACN. A mixed stock solution of all analytical standards was prepared by dilution of the individual stock solutions to a concentration of 10 $\mu$g$\cdot$mL$^{-1}$ in ACN. The stock solutions were stored at -18 °C. For the external calibration standards, the mixed stock solution was freshly diluted to the appropriate concentrations ranging from 2 ng$\cdot$mL$^{-1}$ to 2000 ng$\cdot$mL$^{-1}$ in H2O/ACN 9:1 (v/v). To optimise the SPE procedure for the LOPs, 100 $\mu$L of a 1 $\mu$g$\cdot$mL$^{-1}$ mixed standard solution in H$_2$O/ACN 9:1 (v/v) was added to 20 mL
of a 2 mol$\cdot$L$^{-1}$ sodium chloride solution that was acidified to pH 2 with HCl (30%) to simulate the sample solution after the microwave digestion step.

### 2.2.2 Sampling and preparation of stalagmite samples

Stalagmite *NG01* from the Herbstlabyrinth-Advent-Cave, central Germany, was 50 cm long and had a diameter of approximately 15 cm. It was cut along the growth axis using a diamond blade saw. From one of the two halves, a 1 cm thick slab was
cut, which was then dated using the $^{230}$Th/U-method (Mischel et al., 2016). This showed that the oldest part of the stalagmite grew at ca. 11 000 years BP, whereas the youngest part stems from recent time. Thus, the stalagmite covers the Holocene. The inner part of the stalagmite slab, close to the growth axis, was already used for stable isotope and trace element (Mischel et al., 2016, 2017) as well as fatty acid analysis (Bosle et al., 2014). Thus, the samples for this study had to be taken from the outer part of one half of the stalagmite slab. Pieces of calcite with approximately $0.5-1.2$ cm in width, $2.5-3.7$ cm in length and a
weight of $3.0-5.4$ g were cut from the slab using a diamond wire saw following the growth lines of the stalagmite. Care was taken to always leave 2 cm space to the outer surface of the stalagmite to avoid contamination and dating problems.

To clean the stalagmite samples, each sample was covered with DCM/MeOH 9:1 (v/v) and cleaned for 10 min at 35 °C in an ultrasonic bath. The solvent was discarded, and the cleaning was repeated a second time. Afterwards, the samples were rinsed with ultrapure water, then each sample was covered with ultrapure water, and 250 $\mu$L of HCl (30%) were added to etch away
the outermost layer of calcite, which might be contaminated. After 5 min, the samples were rinsed with ultrapure water, dried and weighed. The samples were then placed in clean glass vials and 2.1 mL of HCl (30%) per gram stalagmite were added to dissolve the calcite over night at room temperature. Before extracting the solutions using SPE, they were diluted 1:1 with ultrapure water to prevent clogging of the cartridges.

### 2.2.3 Sampling and preparation of drip water samples

The drip water samples were collected in the framework of a monthly cave monitoring program (Mischel et al., 2016, 2015). All samples presented here were sampled in October 2015 at different drip sites (two fast drip sites, *D1* and *D5*, with a drip rate





of $0.3 - 0.5$ drops·s$^{-1}$, one slow drip site, *D2*, with a drip rate of approx. 60 mL·month$^{-1}$, and one sample from a cave pool, *PW*). In addition, soil water (*SW*) was sampled in a meadow above the cave, and rain water (*RW*) was sampled at a weather station above the cave. More information on the sampling techniques can be found in (Mischel et al., 2016, 2015). The samples were collected in pre-cleaned glass vessels. To prevent the growth of microorganisms, 5% (w/w) of acetonitrile were added

shortly after sampling. The samples were then stored at 4 °C in the dark for several months. Before extracting the samples using SPE, they were acidified to pH $1 - 2$ with HCl.

### 2.2.4   Preparation of lignin and fresh plant tissue samples

The lignin powder was dissolved in NaOH (2 mol·L$^{-1}$) at a concentration of 1 mg·mL$^{-1}$. 100 μL of this solution was added into the microwave reaction vessel. The plant samples (leaves and branches of Amur maple, and needles and branches of

European yew, all collected in Mainz, Germany) were cut in small pieces and dried in an oven at 50 °C for two days. 10 mg·mL$^{-1}$ were soaked in NaOH (2 mol·L$^{-1}$) for several days. 1 mL of this solution was filtered over 1 $\mu$m filters and added into the microwave reaction vessels.

### 2.2.5   Solid phase extraction of organic matter in dissolved stalagmite solution and drip water samples

The SPE cartridges were preconditioned with 3 mL of MeOH followed by 3 mL of ultrapure water, which was acidified to pH

1-2 with HCl. The diluted stalagmite solution or the acidified drip water sample were loaded onto the cartridges using sample reservoirs. The drip rate was always below 1 drop·s$^{-1}$. The cartridges were washed twice with 3 mL of acidified ultrapure water and dried for 20 min by sucking ambient air through the cartridges using a vacuum manifold. The lignin was eluted with 6 portions of 250 μL of MeOH. The solvent was evaporated to almost dryness under a gentle stream of nitrogen at 30 °C. The residue was re-dissolved in 1.5 mL of NaOH (2 mol·L$^{-1}$), the solution was sonicated for 10 min at 45 °C and added into

the microwave reaction vessel. The sample vial was sonicated again with 1.5 mL of NaOH (2 mol·L$^{-1}$) and this solution was added into the microwave reaction vessel, too.

### 2.2.6   Microwave assisted CuO oxidation

The microwave assisted CuO oxidation procedure was performed according to the method described by Goñi and Montgomery (2000) with slight modifications. An Ethos Plus Microwave Labstation (MLS GmbH, Germany) was used with an HPR-

1000/10S high pressure segment rotor, which can hold up to 10 reaction vessels, and an ATC-CE temperature sensor to measure the temperature inside one reaction vessel. 100 mL Teflon vessels were used as reaction vessels. Each vessel was loaded with 250 mg of CuO, 50 mg of $(NH_4)_2Fe(SO_4)_2 \cdot 6\,H_2O$ and 8 mL of NaOH (2 mol·L$^{-1}$) in total, including the sample solution. The NaOH solution was purged with nitrogen for 30 min before use to remove dissolved oxygen, which could lead to over-oxidation of the lignin oxidation products. For the same reason, the vessels were purged with an argon flow of 1 mL·min$^{-1}$ for 1 min

and then quickly capped to ensure an inert gas atmosphere in the vessels. The vessels were shaken well and then placed in the high-pressure segment rotor of the microwave oven. The temperature was increased to 155 °C in 5 min and then hold at 155



°C for 90 min. Afterwards, the vessels were allowed to cool down to room temperature overnight. Directly after opening the vessels, 50 μL of a 1 μg · mL$^{-1}$ standard solution of ethyl vanillin in H$_2$O/ACN (9:1, v/v) were added as an internal standard into each vessel except the blank sample. The reaction solutions were transferred to 15 mL centrifuge tubes and the reaction vessels were rinsed twice with 3 mL of NaOH (2 mol·L$^{-1}$). The combined solutions were centrifuged for 10 min at 3000 rpm

and the supernatant was decanted into glass vessels. The residue was suspended in 5 mL of NaOH (2 mol·L$^{-1}$) using a vortex mixer, centrifuged again for 10 min at 3000 rpm and the supernatant was combined with the sample solution.

### 2.2.7 UHPLC-ESI-HRMS analysis

The analysis of the lignin oxidation products was carried out on a Dionex Ultimate 3000 ultrahigh-performance liquid chromatography system (UHPLC) that was coupled to a heated electrospray ionisation source (ESI) and a Q-Exactive Orbitrap

high-resolution mass spectrometer (HRMS) (all by Thermo Fisher Scientific, Germany). To separate the LOPs, a Hypersil Gold pentafluorophenyl (PFP) column, 50 mm x 2.1 mm with 1.9 μm particle size (also by Thermo Fisher Scientific, Germany) was used. A H$_2$O/ACN-gradient program was applied. The gradient started with 10% eluent B (consisting of 98% ACN and 2% H$_2$O) and 90% eluent A (consisting of 98% H$_2$O, 2% ACN and 400 μL · L$^{-1}$ formic acid), which was held for 0.5 min. Eluent B was increased to 12% within 2 min, held for 1 min, was further increased to 50% within 1.25 min, held for 0.75

min and increased to 99%. This composition was held for 2 min, then eluent B was decreased to the initial value of 10%.

The ESI source was operated in negative mode, so that deprotonated molecular ions [M-H]$^-$ were formed. The spray voltage was -4.0 kV, the ESI probe was heated to 150 °C to improve the evaporation of the aqueous solvent, the capillary temperature was 350 °C, the sheath gas pressure was 60 psi and the auxiliary gas pressure was 20 psi.

The mass spectrometer was operated in full scan mode with a resolution of 70 000 and a scan range of *m/z* 80 − 500. At the

respective retention time windows, the full scan mode was alternated with a targeted MS$^2$-mode with a resolution of 17 500 to identify the LOPs by their specific daughter ions, see Table 1.

## 3 Results and Discussion

### 3.1 Method development

#### 3.1.1 Separation of LOPs with LC gradient elution and identification of LOPs with MS/MS-experiments

A sufficient separation of the eleven LOPs and two internal standards was achieved within 4.5 min on a PFP column with H$_2$O/ACN-gradient elution, as can be seen in Fig. 1, which shows the normalized chromatogram of 14 LOP standards. The analytes were identified via the exact mass of their molecular ion, their retention time compared to standards and their fragmentation pattern in the MS$^2$ spectrum. As the chromatograms of the real samples were very complex, all three methods were indeed required to identify and quantify the analytes. Whenever possible, the quantification was done by integrating the chro-

matographic peak of the molecular ion. However, when the target analyte peak could not be baseline separated from another signal, the chromatographic peak of a specific daughter ion was used to quantify the analyte.



**Table 1.** Names and abbreviations of the analytes with the respective *m/z* values of their deprotonated molecular ions [M-H]$^-$ and their specific daughter ions.

| name of analyte | abbreviation | *m/z* of [M-H]$^-$ | *m/z* of specific daughter ion (lost neutral fragment) |
|---|---|---|---|
| p-hydroxybenzoic acid | p-Hac | 137.02441 | 93.03455 (-$CO_2$) |
| p-hydroxybenzaldehyde | p-Hal | 121.02943 | 121.02943 (no loss) |
| p-hydroxyacetophenone | p-Hon | 135.04517 | 135.04517 (no loss) |
| vanillic acid | Vac | 167.03498 | 152.01151 (-$CH_3$) |
| vanillin | Val | 151.04007 | 136.01657 (-$CH_3$) |
| acetovanillone | Von | 165.05572 | 150.03220 (-$CH_3$) |
| ethylvanillin (internal standard) | Eval | 165.04518 | 136.01659 (-$CH_2CH_3$) |
| syringic acid | Sac | 197.04555 | 182.02234 (-$CH_3$) |
| syringaldehyde | Sal | 181.05063 | 166.02708 (-$CH_3$) |
| acetosyringone | Son | 195.06628 | 180.04292 (-$CH_3$) |
| trans-ferulic acid | t-Fac | 193.05063 | 134.03734 (-$CH_3$, -$CO_2$) |
| p-coumaric acid | p-Cac | 163.04007 | 119.05024 (-$CO_2$) |
| trans-cinnamic acid (internal standard) | t-Ciac | 147.04520 | 147.04520 (no loss) |

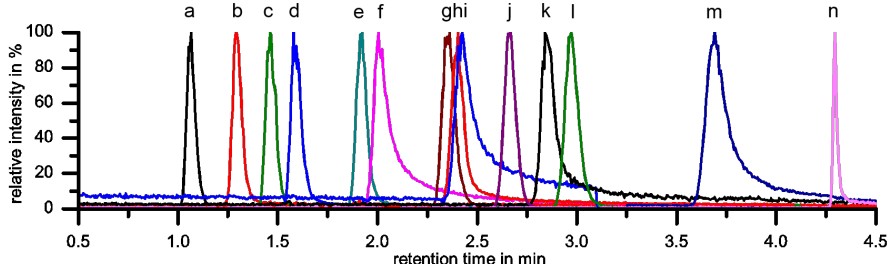

**Figure 1.** Normalized chromatogram of 14 LOP standards on a PFP column. Explanation of the peak numbers (for abbreviations see Table 1): (a) p-Hac, (b) Vac, (c) Sac, (d) p-Hal, (e) p-Hon, (f) Val, (g) p-Cac, (h) Sal, (i) Von, (j) c-Fac, (k) Son, (l) t-Fac, (m) EVal, (n) t-Ciac.

### 3.1.2 Optimisation of the solid phase extraction procedure for LOPs

Two different types of SPE-cartridges were tested. The polymer-based Oasis HLB cartridges (hydrophilic lipophilic balanced polymer, Waters) showed better reproducibility and equal recovery values compared to the silica-based Supelco C18 cartridges (Sigma Aldrich). The recovery rates could be improved by adding ammonia to the elution solvent, ACN or MeOH, as can be seen in Figure 2. The basic pH value of the eluent leads to deprotonation of the phenolic hydroxyl group. In this ionic state,



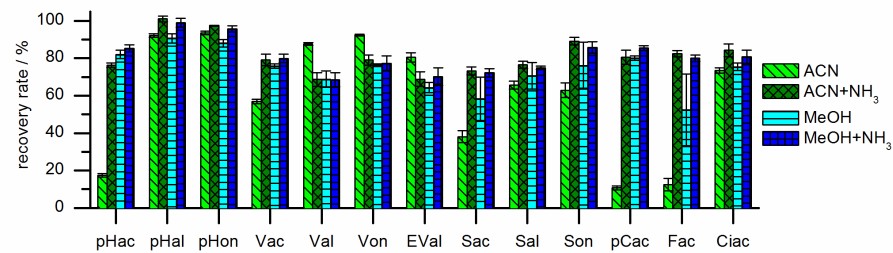

**Figure 2.** Recovery rates of the 13 LOPs on oasis HLB SPE cartridges, eluted with acetonitrile (ACN), acetonitrile with ammonia (ACN+NH$_3$), methanol (MeOH) and methanol with Ammonia (MeOH+NH$_3$). The recovery rates improved significantly if ammonia was added to the elution solvent.

the analytes are better soluble in the polar mobile phase and their adsorption to the stationary phase is weakened. Since we observed an oxidation of aldehydes and an isomerisation of p-coumaric acid and ferulic acid when MeOH was used as elution solvent – an observation that has been made before (Lima et al., 2007) – ACN with ammonia was used as elution solvent. The recovery rates ranged from 69% to 101% and are shown in Fig. 2 and Table 2.

Ethyl acetate was tested as elution solvent, too, as used by Kögel and Bochter; however, the recovery rates were lower than with methanol or acetonitrile. In addition, it was observed that with ethyl acetate, aldehydes were lost in the evaporation step (Fig. A1 in the appendix A1). The SPE method was tested with spikes of LOP standards of different concentrations reaching from 25 ng to 1000 ng. The recovery rate was constant at all concentration levels and the linearity was very good (R$^2$ > 0.9990) for all analytes (Fig. A2 and A3 in the appendix).

**3.1.3   Comparison of different durations and temperatures of the CuO oxidation method**

In former studies, the duration of the CuO oxidation method varied between 90 min and 180 min and temperatures of 150 °C or 170 °C have been applied. Therefore, we compared temperatures of 155 °C and 175 °C (the temperature of the microwave program was chosen 5 °C higher than the desired temperature in the Teflon vessels) and durations of 90 min and 180 min, using 100 µg of mixed lignin as standard sample and three subsamples for each constellation. The results are shown in Fig. 3.

At a temperature of 175 °C and a duration of 180 min, the concentrations of almost all LOPs were dramatically diminished, probably due to overoxidation. For Val, Von, Sal, pCac and Fac, the highest concentrations were reached with 155 °C and 90 min, every increase in temperature or duration of the oxidation step resulted in a loss of analyte. In consequence, the C/V ratio decreased from 0.037 for 155 °C, 90 min to 0.018 for increased temperature, to 0.014 for increased duration and to 0.009 if both were increased. Similarly, the Vac/Val ratio increased from 0.44 for 155 °C, 90 min to 0.83 for increased temperature and to

0.54 for increased duration. For the Sac/Sal ratio, the increase was from 0.16 for 155 °C, 90 min to 0.37 and 0.28, respectively. These results show, that especially the C-group LOPs, pCac and Fac, as well as the aldehydes Val and Sal and the ketone Von are prone to overoxidation. Therefore, care should be taken to adjust temperature and duration of the CuO oxidation step to avoid overoxidation of the LOPs, otherwise the lignin oxidation parameters C/V, S/V and acid/aldehyde ratios will be distorted.



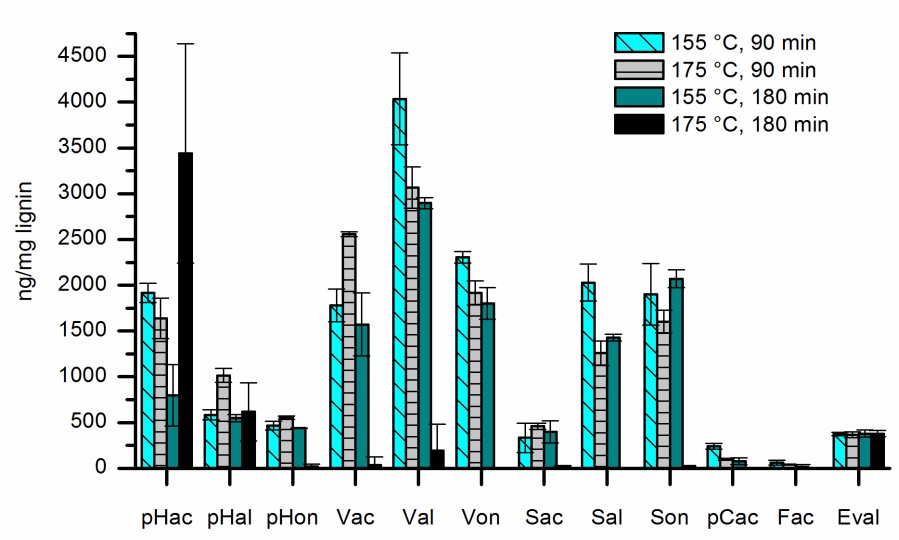

**Figure 3.** Results of a CuO oxidation step at 155 °C and 90 min (light blue bars with diagonal stripes), 175 °C and 90 min (grey bars with horizontal stripes), 155 °C and 180 min (dark cyan bars) and 175 °C and 180 min (black bars). Eval was added after the CuO oxidation step as internal standard

The prevention of overoxidation by the addition of glucose was also tested; however, this did not improve the analysis (see Fig. A4 and Fig. A5 in the SI).

### 3.1.4 Comparison of two sample preparation methods – acid digestion of the stalagmite samples and direct CuO oxidation of stalagmite powder

5   Obviously, each individual step in the analytical sample preparation method includes the risk of positive or negative artefacts, especially if large amounts of chemicals are added. Therefore, experiments were performed to test whether the HCl dissolving step can be skipped by grinding the stalagmite sample and directly adding the powder into the microwave reaction vessels. 24 g of cleaned stalagmite sample were coarsely crushed and mixed. 12 g of this sample mixture were dissolved in HCl and extracted via SPE as described above. The solution was then divided into three subsamples. The other 12 g were finely ground

10  in a mortar, divided into three subsamples and added directly into the microwave reaction vessels. Figure 4 shows that the LOP concentrations found in the acid digested samples were higher for most analytes than in the ground samples. An explanation for this finding might be that at least a part of the lignin particles is bound in the calcite crystals and is only fully released in the acid digestion method. Blyth et al. already stated similar findings for lipid biomarkers (Blyth et al., 2006). Consequently, the acid digestion step is essential for the analysis of the target analytes in speleothems.




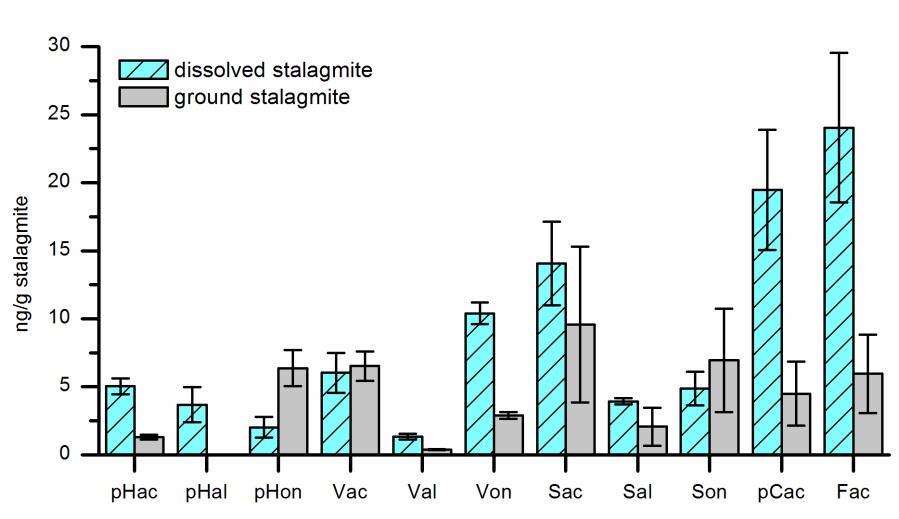

**Figure 4.** Results of dissolved stalagmite samples (light blue bars with diagonal stripes) compared to ground stalagmite samples (grey bars). Dissolving of the samples lead to higher amounts of LOPs.

## 3.2  Method validation

### 3.2.1  Procedural blanks and limits of detection and quantification

To eliminate the influence of possible contamination sources on the results, a procedural blank, which had undergone all sample preparations steps, was analysed with every batch of samples. The concentrations of LOPs measured in this blank were

5   subtracted from the concentrations measured in the samples. The mean values of six procedural blanks measured on different days are shown in Table 2. The values ranged from 0.2 ng to 136 ng, depending on the analyte (see also 3.2.2). The blank value varied from batch to batch, which is reflected in the standard deviations of the blank values given in Table 2. Therefore, the limit of detection (LOD) and the limit of quantification (LOQ) were calculated by using formula (1) and (2), after subtraction of the blank value. The LOD was below 2.5 ng for all analytes and the LOQ was below 15 ng for all analytes.

10   $$\text{LOD} = \frac{3 \cdot \sigma - b}{m} \qquad \text{with } \sigma = \text{standard deviation of the blank value, } b = \text{intersect with the Y-axis and } m = \text{slope} \qquad (1)$$

$$\text{LOQ} = \frac{10 \cdot \sigma - b}{m} \qquad (2)$$



**Table 2.** Limit of detection after blank subtraction (LOD), limit of quantification after blank subtraction (LOQ), mean value of three subsamples of 3.4 g stalagmite after blank subtraction, mean blank value of six procedural blanks measured on different days, and recovery values of the SPE procedure to extract LOPs (Recov. SPE). For the methods of calculation used please refer to the text. The abbreviations for the analytes are shown in Table 1.

| analyte | LOD/ng | LOQ/ng | Mean stalagmite/ng (n=3) | Mean blank/ng (n=6) | Recov. SPE/% (n=3) |
|---|---|---|---|---|---|
| p-Hac | 2.3 | 8.2 | $10 \pm 6$ | $31 \pm 26$ | $76 \pm 1$ |
| p-Hal | 2.7 | 13.7 | $5 \pm 17$ | $136 \pm 66$ | $101 \pm 2$ |
| p-Hon | 0.0 | 0.8 | $11 \pm 1$ | $16 \pm 4$ | $97 \pm 0$ |
| Vac | 2.4 | 8.2 | $66 \pm 16$ | $12 \pm 6$ | $79 \pm 3$ |
| Val | 1.2 | 4.7 | $0 \pm 4$ | $13 \pm 6$ | $69 \pm 4$ |
| Von | 0.9 | 2.5 | $281 \pm 28$ | $4 \pm 2$ | $79 \pm 3$ |
| Sac | 0.5 | 0.6 | $28 \pm 2$ | $1.2 \pm 0.6$ | $73 \pm 2$ |
| Sal | 0.9 | 1.9 | $2.6 \pm 1.3$ | $1.0 \pm 0.8$ | $77 \pm 2$ |
| Son | 0.3 | 1.4 | $22 \pm 6$ | $0.8 \pm 0.8$ | $89 \pm 2$ |
| t-Fac | 0.5 | 0.6 | $20 \pm 0.1$ | $1.0 \pm 0.4$ | $83 \pm 2$ |
| p-Cac | 0.0 | 0.3 | $39 \pm 12$ | $89 \pm 101$ | $81 \pm 4$ |
| Eval (IS) | 0.1 | 0.4 | $29.4 \pm 0.8$ | $0.2 \pm 0.2$ | $69 \pm 4$ |
| Ciac | 0.3 | 1.9 | $21 \pm 7$ | $20 \pm 4$ | $84 \pm 3$ |

### 3.2.2 Blank values

The blank values shown in Table 2 reflect the natural occurrence of the different analytes. The highest blank values have been found for the p-hydroxy group, p-coumaric acid, cinnamic acid, vanillin and vanillic acid. The p-hydroxy group is known to originate not only from lignin, but also from protein rich material such as bacteria (Jex et al., 2014). For p-hydroxy acetophenone, which has a lower blank value than p-hydroxy benzoic acid and p-hydroxy benzaldehyde, it is in discussion whether it originates from lignin or from other sources (Dittmar and Lara, 2001). P-coumaric acid occurs in sporopollenin (Fraser et al., 2012; Montgomery et al., 2016), which is a major component of pollen and fungal spores and also occurs in some form of algae (Delwiche et al., 1989). Therefore, para-coumaric acid might be introduced into the sample via the laboratory air or via insufficiently purified water. Vanillin and its oxidised form vanillic acid are frequently used as perfumes and flavourings in food, cosmetics and household cleaning products. Therefore, these compounds might also be introduced into the sample via the laboratory air or via detergents used to clean the lab ware. Cinnamic acid is used as a perfume and flavouring, too, and it also occurs naturally in bacteria, fungi and algae, as it is part of the shikimate pathway (Dewick, 2009). In this study, cinnamic acid was found in the blank and in all samples. Therefore, cinnamic acid is not suitable as internal standard in the analysis of LOPs in natural samples, although it has been used as internal standard in many studies before (Goñi and Montgomery, 2000;





Kaiser and Benner, 2012). Ethyl vanillin is much more suitable as internal standard, because, as an artificial compound, it has very low blank values and does not occur in natural samples.

### 3.2.3 Reproducibility

10.2 g stalagmite were dissolved, and the solution divided into three subsamples containing 3.4 g stalagmite to determine the
reproducibility of the sample preparation and analysis method. The mean values and standard deviations for all analytes are shown in Table 2. The relative standard deviations ranged from 0.7% to 32% for analytes with more than 2.6 ng (50% for Sal with $2.6 \pm 1.3$ ng). For the p-hydroxy group, the relative standard deviations were higher, but these analytes were not used for the determination of LOP parameters. The LOP parameters calculated from these three subsamples were a C/V ratio of $0.17 \pm 0.04$ and an S/V ratio of $0.15 \pm 0.02$. The variability was mainly caused by the CuO oxidation step, which is known
to cause relatively high variability even in samples with higher lignin content (for example Hedges and Mann (1979) with standard deviations ranging between 3% and more than 80%). The SPE method used for the extraction of LOPs had standard deviations between 1–6% (Table 2) and therefore did not contribute much to the overall variability of the method.

### 3.3 Application to real samples

### 3.3.1 Analysis of plant and lignin samples

The method was applied to different natural samples from known sources to verify that the C/V and S/V ratios are in accordance with published values. The results are shown in Table 3, and their S/V and C/V ratios are visualised in Figure 5. As expected, the highest concentrations of LOPs are found in the lignin from conifer wood with a $\sigma 8$ value of 75.76 $\mu g \cdot mg^{-1}$ as well as in the lignin from wheat straw and mixed wood with a $\Sigma 8$ value of 14.16 $\mu g \cdot mg^{-1}$. This means that the CuO oxidation method has a conversion factor of 1.4–7.6% (w/w) if applied to pure lignin, and that the conversion factor also depends on the type of
lignin. The plant tissue samples gave LOP concentrations ($\Sigma 8$) of 2.3–6.8 $\mu g \cdot mg^{-1}$ for the wood and bark samples and 1.24–1.30 $\mu g \cdot mg^{-1}$ for the leave and needle samples. These concentrations can be explained by the respective lignin content of the different samples. Figure 5 shows the C/V versus S/V diagram for all samples. The regions for different plant types have been defined by Hedges and Mann in 1979 and are based on the analysis of different plant species. Gymnosperm woody samples contain mainly V-group LOPs. Therefore, they plot close to the origin of the diagram. Angiosperm woody samples contain
V- and S-group LOPs, but almost no C-group LOPs. Consequently, they plot close to the S/V-axis. Gymnosperm non-woody samples contain V- and C-group LOPs, but almost no S-group LOPs. Accordingly, they plot close to the C/V-axis. Angiosperm non-woody samples contain all three groups of LOPs and thus show a wide range of C/V and S/V ratios. The analysed plant samples in our study plot all in or close to the expected regions according to their plant type. Only the maple wood and bark sample and the maple leaves sample plot slightly outside of the regions for angiosperm woody and angiosperm non-woody
material, respectively. For the maple wood and bark sample, this could be due to a higher contribution of C-group LOPs in the bark compared to pure woody samples. However, it is important to keep in mind that these regions are just broadly defined and are based on a limited number of analyses and a limited number of different plant species.





**Table 3.** Concentrations of the V, S and C-group LOPs, the sum of all 8 LOPs ($\Sigma 8$) and the ratios C/V and S/V in fresh plant and lignin samples.

| sample | V-group / $\mathrm{mg \cdot g^{-1}}$ | S-group / $\mathrm{mg \cdot g^{-1}}$ | C-group / $\mathrm{mg \cdot g^{-1}}$ | $\Sigma 8$ / $\mathrm{mg \cdot g^{-1}}$ | C/V | S/V |
|---|---|---|---|---|---|---|
| lignin from conifer wood | 75.12±0.77 | 0.293±0.015 | 0.345±0.012 | 75.76±0.77 | 0±0.00 | 0±0.00 |
| lignin from wheat straw and mixed wood | 7.42±0.10 | 6.483±0.078 | 0.255±0.009 | 14.16±0.13 | 0.03±0.00 | 0.87±0.02 |
| yew wood and bark | 2.25±0.04 | 0.024±0.001 | 0.083±0.001 | 2.35±0.04 | 0.04±0.00 | 0.01±0.00 |
| maple wood and bark | 2.87±0.04 | 3.626±0.089 | 0.303±0.002 | 6.80±0.10 | 0.11±0.00 | 1.27±0.04 |
| yew needles | 0.74±0.02 | 0.059±0.001 | 0.494±0.010 | 1.30±0.02 | 0.66±0.02 | 0.08±0.00 |
| maple leaves | 0.75±0.02 | 0.314±0.005 | 0.184±0.003 | 1.24±0.02 | 0.25±0.01 | 0.42±0.01 |

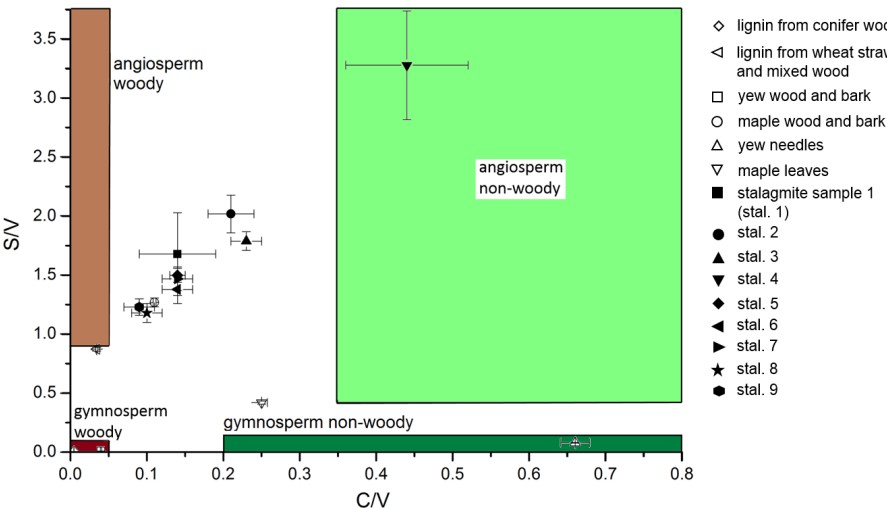

**Figure 5.** Lignin oxidation parameters S/V vs. C/V of different real samples and regions for different sample types defined by Hedges and Mann in 1979.

### 3.3.2 Analysis of stalagmite samples

With a $\Sigma 8$ value of ca. 20–60 $\mathrm{ng \cdot (g\ stalagmite)^{-1}}$ (Table 4), the LOP concentration of the stalagmite samples is five orders of magnitude lower than for the vegetation samples and three to four orders of magnitude lower than the typical concentration of sediment samples (e.g., $\Sigma 8$ is 0.15–0.75 $\mathrm{mg \cdot (g\ sediment)^{-1}}$ in Tareq et al. (2011)). Because of these low concentrations, 3–5 g stalagmite were required for an analysis to be above the limit of quantification. The C/V ratios of the stalagmite samples were all above 0.5, and the S/V ratios were all above 1.0, which suggests a significant contribution of angiosperm woody and angiosperm non-woody vegetation. However, gymnosperm woody and gymnosperm non-woody material might also have




contributed to the lignin pool. This suggests a mixed deciduous forest above the cave, and would be in accordance with the results of Litt et al., who analysed pollen from Holocene lake sediments from the Westeifel Volcanic Field (Litt et al., 2009), which is relatively close to the Herbstlabyrinth.

The nine stalagmite samples were taken at different distances from the top of the stalagmite. This analysis shall serve as

a proof of principle for a higher-resolution analysis of the whole stalagmite. In Figure 6, the C/V and S/V ratios are plotted against distance from top (dft). Both ratios show a pronounced peak at 20 cm dft. Furthermore, both ratios show higher values in the top 15 cm and lower values with a decreasing trend between 30 and 50 cm dft. A higher S/V ratio indicates a higher contribution of angiosperm vegetation to the lignin source, and a higher C/V ratio suggests a higher contribution of non-woody vegetation. Therefore, the peak at 20 cm dft could be interpreted as increased input of non-woody, angiosperm vegetation, such

as grasses, and less input of wood. The decreasing trend in the lower part of the stalagmite indicates a trend towards more woody, gymnosperm vegetation, such as pine forest. Of course, these presumptions have to be proven by a complete analysis of the stalagmite and a comparison with the other proxy data (Mischel et al., 2016). In addition, a comparison with Holocene pollen records from the area may confirm these preliminary results. Overall, these first results show significant variability of the C/V and S/V ratios and, therefore, the lignin sources. This promising result encourages us to use the analysis of LOPs in

stalagmites for paleo vegetation reconstruction.

**Table 4.** Concentrations of the V-, S- and C-group LOPs, the sum of all 8 LOPs ($\Sigma 8$) and the ratios C/V and S/V in stalagmite *NG01* from the Herbstlabyrinth-Advent-Cave.

| sample | V-group / ng $\cdot$ g$^{-1}$ | S-group / ng $\cdot$ g$^{-1}$ | C-group / ng $\cdot$ g$^{-1}$ | $\Sigma 8$ / ng $\cdot$ g$^{-1}$ | C/V | S/V |
|---|---|---|---|---|---|---|
| stalagmite sample 1 | 10.6±1.7 | 20.8±1.7 | 1.7±0.5 | 33.2±2.4 | 0.16±0.05 | 1.96±0.35 |
| stalagmite sample 2 | 13.0±0.5 | 30.3±1.8 | 3.1±0.4 | 46.4±1.9 | 0.24±0.03 | 2.33±0.16 |
| stalagmite sample 3 | 13.0±0.4 | 27.9±0.5 | 3.6±0.3 | 44.5±0.7 | 0.27±0.02 | 2.14±0.08 |
| stalagmite sample 4 | 3.7±0.4 | 14.1±0.7 | 1.9±0.2 | 19.6±0.8 | 0.51±0.08 | 3.84±0.46 |
| stalagmite sample 5 | 28.6±0.8 | 50.4±1.1 | 4.8±0.4 | 83.8±1.4 | 0.17±0.01 | 1.76±0.06 |
| stalagmite sample 6 | 18.2±1.1 | 29.7±1.2 | 3.0±0.3 | 51.0±1.6 | 0.17±0.02 | 1.63±0.12 |
| stalagmite sample 7 | 17.4±0.7 | 29.9±1.3 | 2.7±0.2 | 50.0±1.5 | 0.16±0.02 | 1.72±0.08 |
| stalagmite sample 8 | 17.9±0.7 | 24.8±1.1 | 2.0±0.3 | 44.7±1.3 | 0.11±0.02 | 1.38±0.10 |
| stalagmite sample 9 | 24.8±1.0 | 35.7±0.8 | 2.6±0.4 | 63.1±1.4 | 0.10±0.02 | 1.44±0.07 |

### 3.3.3    Analysis of cave drip water samples

Very little is known about how lignin is transported from the soil into the cave and how it is incorporated into a stalagmite. To gain further understanding about these processes, it is useful to also analyse lignin in cave drip water. The lignin concentration in cave drip water is even lower than in stalagmite samples because crystallisation of calcite also serves as an enrichment step

for the organic components contained in the water. Therefore, a sample volume of 100–200 mL water was used. Here we show





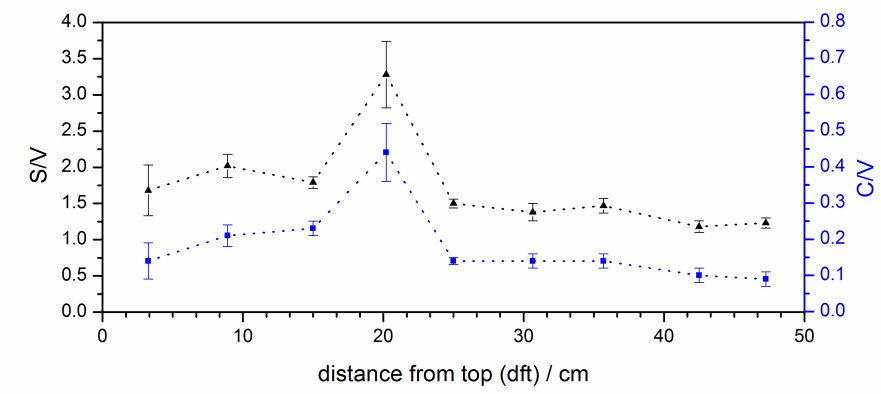

**Figure 6.** S/V (black triangles, left axis) and C/V (blue squares, right axis) ratios of stalagmite *NG01* plotted against the distance from the top of the stalagmite

the results of the analysis of six different water samples from the Herbstlabyrinth-Advent-Cave, all sampled in October 2015 (Table 5). As expected, the soil water (*SW*) has the largest lignin content with almost 2 $\mu g \cdot L^{-1}$. The rain water (*RW*) also has a relatively large lignin content of almost 1.5 $\mu g \cdot L^{-1}$, which is surprising since this water has not been in contact with soil or vegetation. The lignin content of the cave drip water samples is much lower, ranging from 0.2 $\mu g \cdot L^{-1}$ for the pool water

to 0.4 $\mu g \cdot L^{-1}$ for the fast drip site *D1*. Interestingly, the content of the V-group LOPs decreases from soil water to cave drip water by 80–92%, which is much more than the decrease in S-group (70–76%) and C-group LOPs (56–86%) (Fig. 7). This is also reflected in higher S/V and C/V ratios in the cave drip water than in the soil water. For the S/V ratio, an increasing trend from soil water over the two fast drip sites *D1* and *D5* and the slow drip site *D2* to the cave pool water is observed. This could be due to different residence times in the cave and the overlaying karst. These hypotheses should be proven by a further

systematic analysis of cave drip water. This would also enable the study of seasonal variations in the lignin input. The monthly cave monitoring program of Mischel et al. (Mischel et al., 2016, 2015) combined with our new method for the analysis of LOPs even in low-concentration cave drip water could be a valuable tool to further investigate these topics.

## 4    Conclusions and outlook

We developed a sensitive method for the analysis of LOPs in speleothems and cave drip water and tested it successfully

on samples from the Herbstlabyrinth-Advent-Cave. The method was adjusted to the low concentrations of organic matter in speleothems and cave drip water and showed sufficient sensitivity and reproducibility to detect even trace concentrations of lignin. The use of the established CuO oxidation method allows to compare the results to LOP records in other archives. However, as the CuO oxidation step is the main source of variability in our method, an alternative degradation method for lignin with higher reproducibility should be developed. This method could, for example, be based on electrolysis. In addition,

speleothem samples from other caves in different vegetation and climate zones should be analysed to gain more insight into the

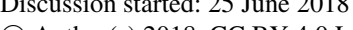



**Table 5.** 5 Concentrations of the V-, S- and C-group LOPs, the sum of all 8 LOPs (Σ8) and the ratios C/V and S/V in different water samples collected at the Herbstlabyrinth-Advent-Cave in October 2015.

| Sample | Sample volume / L | V-group / $ng \cdot L^{-1}$ | S-group / $ng \cdot L^{-1}$ | C-group / $ng \cdot L^{-1}$ | Σ8 / $ng \cdot L^{-1}$ | C/V | S/V |
|---|---|---|---|---|---|---|---|
| RW (rain water) | 0.185 | 918±25 | 345±14 | 192±14 | 1456±33 | 0.21±0.02 | 0.38±0.02 |
| SW (soil water) | 0.076 | 1370±41 | 363±33 | 224±25 | 1956±60 | 0.16±0.02 | 0.26±0.03 |
| D1 (fast dripping) | 0.265 | 271±7 | 87±10 | 81±11 | 439±17 | 0.3±0.04 | 0.32±0.04 |
| D5 (fast dripping) | 0.258 | 175±7 | 95±11 | 99±7 | 369±15 | 0.56±0.05 | 0.54±0.06 |
| D2 (slow dripping) | 0.205 | 157±8 | 107±18 | 81±9 | 346±22 | 0.52±0.06 | 0.68±0.12 |
| PW (pool water) | 0.205 | 114±7 | 88±10 | 32±7 | 234±14 | 0.28±0.07 | 0.77±0.09 |

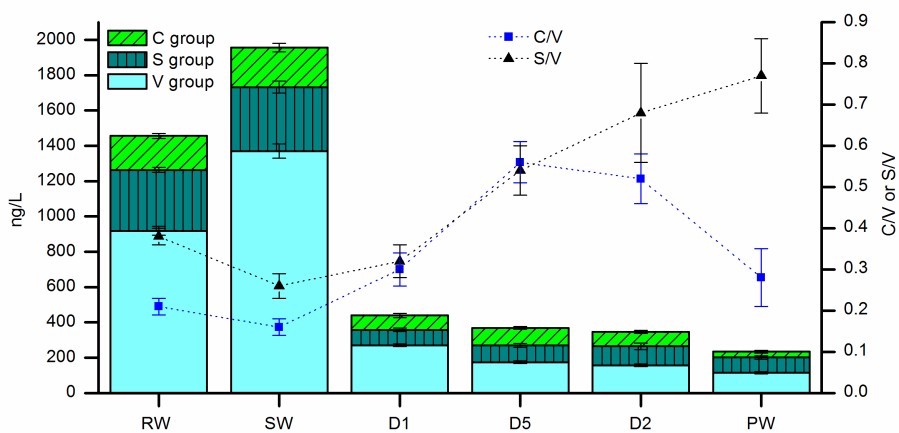

**Figure 7.** LOP concentrations (stacked columns with left axis) and LOP ratios (symbols with right axis) of rain water (*RW*), soil water (*SW*), cave drip water from fast drip sites (*D1* and *D5*), a slow drip site (*D2*) and cave pool water (*PW*). The stacked columns contain the V-group LOPs (light cyan bars), S-group LOPs (dark cyan bars with vertical stripes) and C-group LOPs (green bars with diagonal stripes). Black triangles show the S/V ratio and blue squares show the C/V ratio.

relation of vegetation, climate and the LOP signal in speleothems. The analysis of cave drip water, sampled monthly within the framework of a cave monitoring program, could elucidate seasonal variations of lignin input as well as possible fractionation processes during its pathway from the soil to the cave.



## Appendix A: Supplementary Information

### A1 Evaporation effects of different elution solvents for SPE

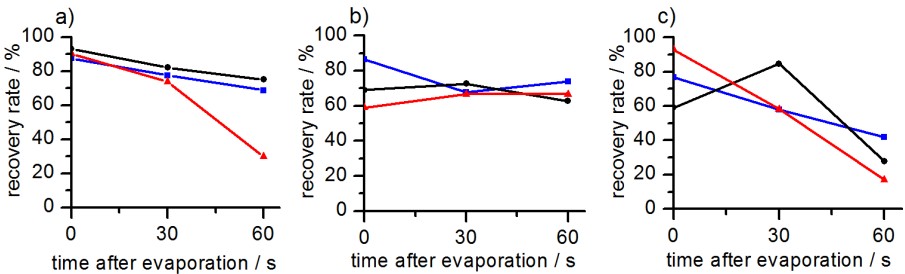

**Figure A1.** Recovery rates of vanillin after evaporation in a) acetonitrile, b) methanol and c) ethyl acetate at 45 °C (red triangles), 30 °C (black circles) and 25 °C (blue squares). The residue was re-dissolved in $H_2O$/ACN 9:1 (v/v) and analysed. At elevated evaporation temperatures, vanillin and other aldehydes evaporated and were lost for analysis. In ethyl acetate, this evaporative loss was more pronounced than in acetonitrile and methanol.

### A2 Linearity test of the SPE cartridges at different spiking concentrations

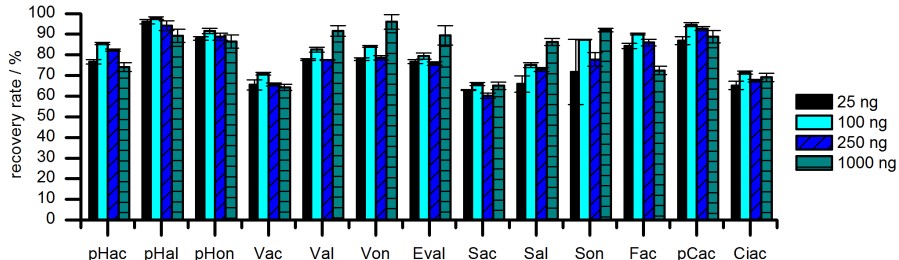

**Figure A2.** Recovery rates of the solid phase extraction of LOPs at different spiking concentrations from 25 ng to 1000 ng. (For information on the surrogate solution see figure caption of Fig. A3.)





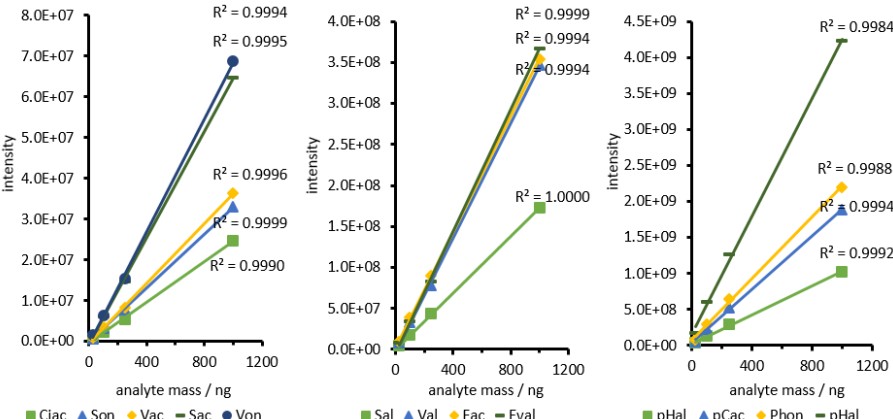

**Figure A3.** Linearity test of the SPE method for the extraction of LOPs. 20 mL of a surrogate sample solution ($2 \, \mathrm{mol \cdot L^{-1}}$ NaCl in ultrapure water, acidified with HCl to pH 2) were spiked with 25 ng, 100 ng, 250 ng and 1000 ng of LOP standards.

### A3 Test of the addition of glucose to prevent overoxidation

Many studies (e.g., Kaiser and Benner, 2012; Louchouarn et al., 2000; Spencer et al., 2010) recommended to add glucose to samples with low organic carbon content to prevent overoxidation of aldehydes. As stalagmite samples do have a low organic carbon content compared to soil or sediment samples, we tested the addition of glucose. The result was that the ratio

5    of Vac/Val did indeed decrease from 0.48±0.11 without glucose to 0.26±0.12 with glucose, because the yield of vanillin increased with the addition of glucose. Nevertheless, the ratio of C/V decreased from 0.46±0.12 to 0.26±0.31 and the ratio of S/V decreased from 0.76±0.19 to 0.41±0.36 (Fig. A4). This means that the addition of glucose did not prevent cinnamyl and syringyl phenols from overoxidation. In contrast, there were more interfering peaks in the chromatograms with glucose (Fig. A5), which made integration difficult and lead to increased uncertainty in quantification. Consequently, no glucose was added

10    in the CuO oxidation step.



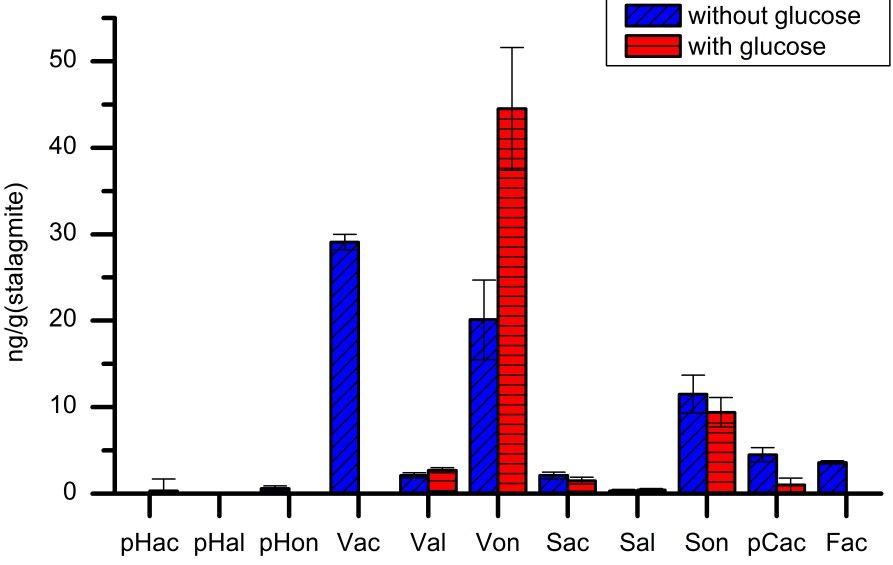

**Figure A4.** Comparison of LOP concentrations with and without the addition of glucose. Only for Von, there was an increase in the concentration with the addition of glucose. For all other analytes, the method without glucose gave better results.

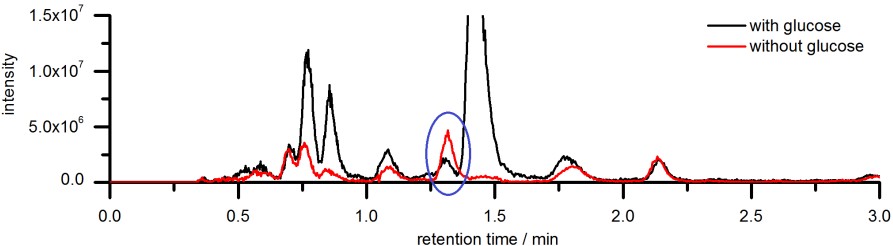

**Figure A5.** Chromatogram of *m/z* 167.03498 (vanillic acid) with (black line) and without (red line) the addition of glucose. The peak of vanillic acid is circled. It was higher and better separated from neighbouring peaks without the addition of glucose. Similar observations were made for other analytes, too.



*Competing interests.* The authors declare that they have no conflict of interest.

*Acknowledgements.* We thank Dr. Simon Mischel for providing stalagmite and cave drip water samples from the Herbstlabyrinth-Advent-Cave. This project has received funding from the European Union's Horizon 2020 research and innovation programme under the Marie Skłodowska-Curie grant agreement No 691037. Denis Scholz acknowledges funding of the German Research Foundation (SCHO 1274/3-1

5  and SCHO 1274/9-1)




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
