# Peer review of "Quantification of lignin oxidation products as vegetation biomarkers in speleothems and cave drip water"

_Biogeosciences, 2018_

## Referee Comment (RC1) · Anonymous Referee #1 · 2 Jul 2018

After reading the manuscript I have the following remarks and suggestions. 1.Elements of scientific novelty should be presented in a more detailed and convincing manner (in the last paragraph of the Introduction). 2. I suggest that a diagram presenting the steps of the procedure used in the study be added to the EXPERIMENTAL section. It would help understand the details of the analytical protocol better, and allow the written description of the procedure to be shortened. 3. Innovative potential of the results obtained should be explained in detail (CONCLUSIONS) 4.Application of proper quality assurance/quality control (QA/QC) procedures is vital for the measurement results to be treated as a source of reliable analytical information. Consequently, I suggest that a separate section devoted to QA/QC be added to the manuscript. Spe-

cial attention should be paid to: - description of the validation procedure for the applied/proposed analytical protocol, - information on metrological characteristics of the analytical procedure, especially Method Quantitation Limit (MQL) values for the entire procedure (from handling of representative samples to statistical and chemometric evaluation of the data sets obtained), and not only for the analytical techniques used during the analysis of the extracts. 5. I suggest that the protocol described in Journal of Chromatography A (1217, 882-891, 2010) entitled "Estimating uncertainty in analytical procedures based on chromatographic techniques" can be used for evaluation and calculation of expanded uncertainty of results obtained when the procedure described in this manuscript is applied. 6. Green aspects of different approaches known from the literature should be discussed. There is a strong need of insertion of an additional chapter to the text of the paper. In this paper the newest literature information on the development of green analytical principles and approaches should be presented. Green analytical Chemistry (GAC) should be treated as a very important part of green chemistry. Authors should study the literature data this field in deeper manner.

---

## Referee Comment (RC2) · Anonymous Referee #2 · 21 Jul 2018

The manuscript from Heidke et al. proposes a sensitive method to quantify lignin markers in speleothems and cave drip water with application to real samples. The paper is of good quality and of interest for the readers of Biogeosciences. I have only minor comments.

I agree with reviewer #1 concerning the suggestion to add a QA/QC section and a diagram/scheme of the sample preparation protocol.

I suggest using consistent unit measures when reporting concentration values throughout the manuscript, especially concerning Table 2, Table 3 and section 3.3.1. I would also report LODs and LOQs in terms of concentrations rather than absolute amounts

as it may be ambiguous if the absolute amount is referred to the total amount of samples or the total amount of analyte injected in the instrument.

I would suggest removing the first paragraph of the introduction or, alternatively, combined it with the last paragraph of the introduction.

At line 6 of page 5, the authors state that samples were stored for several months. Was the conservation of the samples tested somehow?

Section 2.2.7, please add the injection volume for analysis and the settings for MS/MS (e.g. collision method and energies).

I suggest renaming section 3.2.3 "repeatability" as, if I have understood correctly, it describes repetitions with the same equipment and the same operator.

Please add the total volume of the surrogate solutions to the caption of figure A2.

Typos: Page 3, line 32: $H_2O$ Page 12, line 17: the symbol sigma should be capitalised

---

## Author Comment (AC1) · 24 Aug 2018

We thank the two anonymous reviewers for carefully evaluating our manuscript. Our point-bypoint reply directly follow the referee comments (blue) and appears in black after each comment. When we refer to text passages of the manuscript we use quotation marks and *cursive font*, new or altered text segments are printed in green.

**Response to Referee #1**

**Referee comment 1:**

1.Elements of scientific novelty should be presented in a more detailed and convincing manner (in the last paragraph of the Introduction).

**Response 1:**

We have rewritten the last paragraph of the introduction to better point out the scientific novelty of our work. The last paragraph of the introduction now reads:

"The purpose of this study was to develop and validate a sensitive and selective method for the quantification of LOPs in both speleothem and cave drip water samples using liquid chromatography electrospray ionisation mass spectrometry (LC-ESI-MS). This method offers new possibilities for paleo-vegetation reconstruction since it combines the advantages of lignin analysis as a highly specific vegetation biomarker with the above-mentioned benefits of speleothems as unique terrestrial climate archives. Lignin as a vegetation biomarker is much more specific for higher plants than for example n-alkanes or fatty acids (Jex et al., 2014), and thus can help to interpret other vegetation markers and stable isotope records. Up to now, lignin analysis for paleo vegetation reconstruction has only been applied to lake sediments and peat cores, which contain much larger amounts of organic matter than speleothems. Our method allows to analyse the lignin composition of trace amounts of organic matter preserved in speleothems. The stalagmite samples are first acid digested, and the acidic solution is then extracted by SPE. The eluent is then subjected to CuO oxidation in a microwave assisted digestion method. The oxidised sample solutions are again extracted and enriched by SPE, and the LOPs are then separated and detected by ultrahigh-performance liquid chromatography coupled to electrospray ionisation high-resolution mass spectrometry (UHPLC-ESI-HRMS)."

**Referee comment 2:**

I suggest that a diagram presenting the steps of the procedure used in the study be added to the EXPERIMENTAL section. It would help understand the details of the analytical protocol better, and allow the written description of the procedure to be shortened.

**Response 2:**

We thank the reviewer for this helpful suggestion. We added a diagram of the analytical procedure, which is shown below. However, we decided not to shorten the written description of the individual steps of the procedure, because we think that all given details are necessary for the reader to be able to reproduce the analytical method.

**Figure 1.** Process chart of the overall sample preparation procedure. A detailed description of the individual steps is given in section 2.2.

**Referee comment 3:**

Innovative potential of the results obtained should be explained in detail (CONCLUSIONS).

**Response 3:**

We have rewritten the conclusion and we now write:

**"Conclusion and outlook**

We developed a sensitive method for the quantification of LOPs in speleothems and cave drip water and tested it successfully on samples from the Herbstlabyrinth-Advent-Cave. This is, to our knowledge, the first quantitative analysis of LOPs in speleothems and cave drip water. Our method provides a new and highly specific vegetation proxy for the reconstruction of paleo vegetation and paleo climate from speleothem archives. The method was adjusted to the low concentrations of organic matter in speleothems and cave drip water and showed sufficient sensitivity to detect even trace concentrations of lignin. The use of the established CuO oxidation method allows to compare the results to LOP records in other archives. However, as the CuO oxidation step is the main source of variability in our method, an alternative degradation method for lignin with higher reproducibility should be developed. [...]"

**Referee comment 4:**

Application of proper quality assurance/quality control (QA/QC) procedures is vital for the measurement results to be treated as a source of reliable analytical information. Consequently, I suggest that a separate section devoted to QA/QC be added to the manuscript. Special attention should be paid to: - description of the validation procedure for the applied/proposed analytical protocol, - information on metrological characteristics of the analytical procedure, especially Method Quantitation Limit (MQL) values for the entire procedure (from handling of representative samples to statistical and chemometric evaluation of the data sets obtained), and not only for the analytical techniques used during the analysis of the extracts.

**Response 4:**

We thank the reviewer for this helpful comment. We have revised section 3.2 Method validation and expanded it to a QA/QC section. We now write:

**"3.2 Method validation and Quality assurance**

**3.2.1 Selectivity**

The selectivity of the method was assured by using three parameters for peak identification: the retention time, the exact m/z ratio of the analyte, and the  $MS^2$ -spectra, as described in section 3.1.1. The variation in the retention time was  $\pm$  0.01 min. To assure that the measured peak area was caused only by the analyte, the corresponding peak area of the reagent blank measurement was subtracted.

**3.2.2 Calibration and linearity**

External calibration with a standard mixture containing all analytes was performed. The calibration function was obtained using the linear regression method. The parameters of the individual calibration functions are shown in Table A1 in the supplementary information. The concentrations of the standards ranged from  $20-500 \text{ ng} \cdot \text{mL}^{-1}$  for stalagmite and drip water samples and up to  $2000 \text{ ng} \cdot \text{mL}^{-1}$  for plant and lignin samples. The calibration was linear in this range.

**3.2.3 Limits of detection and quantification and reagent blanks**

[revised manuscript text omitted]

| analvte   | MDL / MOL /         |                     | Mean stalag-
mite /   | Mean stalag-
mite /   | Mean blank / Recov. SPF / |         |  |
|-----------|---------------------|---------------------|--------------------------|--------------------------|---------------------------|---------|--|
| unury ce  | ng∙mL −1 | ng∙mL -1 | $ng \cdot mL^{-1}$ (n=3) | ng·g -1 (n=3) | ng·mL −1 (n=6) | % (n=3) |  |
| р-Нас     | 13.8                | 41.9                | 50 ± 30                  | 2.9 ± 1.8                | 155 ± 130                 | 76 ± 1  |  |
| p-Hal     | 25.9                | 78.4                | 25 ± 85                  | 1.5 ± 5.0                | 680 ± 330                 | 101 ± 2 |  |
| p-Hon     | 2.3                 | 7.0                 | 55 ± 5                   | 3.2 ± 0.3                | 80 ± 20                   | 97 ± 0  |  |
| Vac       | 13.7                | 41.5                | 330 ± 80                 | 19.4 ± 4.7               | 60 ± 30                   | 79 ± 3  |  |
| Val       | 8.2                 | 24.8                | 0 ± 20                   | 0.0 ± 1.2                | 65 ± 30                   | 69 ± 4  |  |
| Von       | 3.7                 | 11.3                | 1405 ± 140               | 82.6 ± 8.2               | 20 ± 10                   | 79 ± 3  |  |
| Sac       | 0.3                 | 0.8                 | 140 ± 10                 | 8.2 ± 0.6                | 6 ± 3                     | 73 ± 2  |  |
| Sal       | 2.3                 | 7.1                 | 13 ± 6.5                 | 0.8 ± 0.4                | 5 ± 4                     | 77 ± 2  |  |
| Son       | 2.5                 | 7.7                 | 110 ± 30                 | 6.5 ± 1.8                | 4 ± 4                     | 89 ± 2  |  |
| t-Fac     | 2.0                 | 6.2                 | 100 ± 0.5         | 5.9 ± 0.0                | 5 ± 2                     | 83 ± 2  |  |
| p-Cac     | 0.2                 | 0.7                 | 195 ± 60                 | 11.5 ± 3.5               | 445 ± 505                 | 81 ± 4  |  |
| Eval (IS) | 0.6                 | 1.8                 | 147 ± 4                  | 8.6 ± 0.2                | 1 ± 1                     | 69 ± 4  |  |
| Ciac      | 3.8                 | 11.6                | 105 ± 35                 | 6.2 ± 2.1                | 100 ± 20                  | 84 ± 3  |  |

**3.2.5 Repeatability**

To determine the repeatability of the sample preparation and analysis method, 10.2 g stalagmite were dissolved, and the solution divided into three subsamples containing 3.4 g stalagmite. The mean values and standard deviations for all analytes are shown in Table 2. The relative standard deviations ranged from 0.7% to 32% for analytes with more than 2.6 ng (50% for Sal with 2.6±1.3 ng). For the p-hydroxy group, the relative standard deviations were higher, but these analytes were not used for the determination of LOP parameters. The LOP parameters calculated from these three subsamples were a C/V ratio of 0.17±0.04 and an S/V ratio of 0.15±0.02. The variability was mainly caused by the CuO oxidation step, which is known to cause relatively high variability even in samples with higher lignin content (for example Hedges and Mann (1979) with standard deviations ranging between 3% and more than 80%). The SPE method used for the extraction of LOPs had standard deviations between 1–6% (Table 2) and therefore did not contribute much to the overall variability of the method.

**3.2.6 Estimation of uncertainty"**

... see Response 5

**Referee comment 5:**

I suggest that the protocol described in Journal of Chromatography A (1217, 882-891, 2010) entitled "Estimating uncertainty in analytical procedures based on chromatographic techniques" can be used for evaluation and calculation of expanded uncertainty of results obtained when the procedure described in this manuscript is applied.

Response 5:

We thank the reviewer for this helpful suggestion. We have applied the suggested protocol and have added a subsection on the estimation of uncertainty in section *3.2 Method validation and quality assurance*. We now write:

**"3.2.6 Estimation of uncertainty**

According to Konieczka and Namiesnik (2010), the main factors contributing to the uncertainty budget are the uncertainty of the measurement of the weight or volume of the sample,  $u_r$ (sample), the repeatability of the sample preparation procedure,  $u_r$ (rep.), the recovery determination of the internal standard,  $u_r$ (recov.), the calibration step,  $u_r$ (cal.), and the uncertainty associated with analyte concentrations close to the limit of detection,  $u_r$ (LOD). The combined relative uncertainty  $U_r$  is expressed in equation 5.

$$U_{r} = \sqrt{(u_{r}(sample))^{2} + (u_{r}(rep.))^{2} + (u_{r}(recov.))^{2} + (u_{r}(cal.))^{2} + (u_{r}(LOD))^{2}}$$
(5)

In our method,  $u_r$ (sample) is relatively small with 1 mg or 1 mL, which is usually < 1%. The uncertainty associated with the repeatability of the sample preparation, calculated as the standard deviation of three individually prepared subsamples as explained in section 3.2.5, has the largest influence and can equal 1–30%. The uncertainty of the recovery determination of the internal standard, calculated as the standard deviation of the internal standard, contributes with 1–6%.  $u_r$ (cal.), calculated as the standard deviation of the concentration determination of three injections of the same sample into the LC-MS-system, can equal 1–15%, but is usually around 3–5%.  $u_r$ (LOD), calculated according to equation (6), depends strongly on the concentration c of the analyte.

$$u_r = \frac{LOD}{c} \tag{6}$$

In the data for stalagmite samples presented in Table 2,  $u_r(LOD)$  equals 0.1–5% for most analytes, 17% for Sal and 27–100% for the p-hydroxy group.

The errors for all results presented in this work were calculated using the law of propagation of uncertainty. All equations used for calculating concentrations, lignin oxidation parameters and errors are shown in section A4 in the supplementary information."

**Referee comment 6:**

Green aspects of different approaches known from the literature should be discussed. There is a strong need of insertation of an additional chapter to the text of the paper. In this paper the newest literature information on the development of green analytical principles and approaches should be presented. Green analytical Chemistry (GAC) should be treated as a very important part of green chemistry. Authors should study the literature data this field in deeper manner.

**Response 6:**

We thank the reviewer for this interesting and helpful comment. We have studied the recent literature on Green analytical Chemistry and added an additional section about this topic. We now write:

**"Aspects of green analytical chemistry**

When developing a new analytical method, it is advantageous to consider how environmentfriendly (green) the different approaches are. The principles of green analytical chemistry include, among others, to generate as little waste as possible, to eliminate or replace toxic reagents, to miniaturize analytical instruments or to avoid derivatization (Gałuszka et al., 2013; Armenta et al., 2008). In our method, we tried to favour greener approaches over less green approaches whenever possible without sacrificing other qualities like sensitivity. We used solid phase extraction, which consumes considerably less solvent than liquid-liquid extraction, and UHPLC, which is less solvent and time consuming than HPLC. In addition, liquid chromatography does not require a derivatization step, as opposed to gas chromatography. However, the least green step in our method is the CuO oxidation step, as it generates toxic waste and consumes energy. We still chose the CuO oxidation method for our proof of principle analysis because it is the most widely used lignin degradation method for the analysis of LOPs and therefore allows us to compare our results with existing LOP records. In the future, however, a greener approach to the degradation of lignin to LOPs should be chosen, which could, for example, be based on electrolysis, preferably in a miniaturized flow cell (Leppla, 2016)."

**Response to Referee #2**

**Referee comment 1:**

I agree with reviewer #1 concerning the suggestion to add a QA/QC section and a diagram/scheme of the sample preparation protocol.

**Response 1:**

As already mentioned above, we have added a QA/QC section, which is described in detail in the response to Referee #1, as well as a diagram/scheme of the sample preparation protocol.

**Referee comment 2:**

I suggest using consistent unit measures when reporting concentration values throughout the manuscript, especially concerning Table 2, Table 3 and section 3.3.1. I would also report LODs and LOQs in terms of concentrations rather than absolute amounts as it may be ambiguous if the absolute amount is referred to the total amount of samples or the total amount of analyte injected in the instrument.

**Response 2:**

We thank the reviewer for pointing out this inconsistency in unit measures. We now report all concentrations referring to solid samples (stalagmite, plant samples, lignin powder) in ng/g,  $\mu$ g/g or mg/g, all concentrations referring to drip water samples in ng/L, and all concentrations referring to the final sample solution injected into the instrument (standards, blanks, LODs, LOQs) in ng/mL. In case of ambiguity, we specify what the concentration refers to. We do not use absolute amounts anymore.

**Referee comment 3:**

I would suggest removing the first paragraph of the introduction or, alternatively, combined it with the last paragraph of the introduction.

**Response 3:**

We have removed the first paragraph of the introduction and have combined it with the last paragraph of the introduction. The introduction now starts directly with the sentence:

**"Speleothems are calcareous mineral deposits that form within caves in karstified carbonate rock. The most common types of speleothems are...".**

The last paragraph of the introduction now reads:

"The purpose of this study was to develop and validate a sensitive and selective method for the quantification of LOPs in both speleothem and cave drip water samples using liquid chromatography electrospray ionisation mass spectrometry (LC-ESI-MS). This method offers new possibilities for paleo vegetation reconstruction since it combines the advantages of lignin analysis as a highly specific vegetation biomarker with the above-mentioned benefits of speleothems as unique terrestrial climate archives. Lignin as a vegetation biomarker is much more specific for higher plants than for example n-alkanes or fatty acids (Jex et al., 2014), and thus can help to interpret other vegetation markers and stable isotope records. Up to now, lignin analysis for paleo vegetation reconstruction was only applied to lake sediments and peat cores, which contain much larger amounts of organic matter than speleothems. Our method allows to analyse the lignin composition of trace amounts of organic matter preserved in speleothems. The stalagmite samples are first acid digested, and the acidic solution is then extracted by SPE. The eluent is then subjected to CuO oxidation in a microwave assisted digestion method. The oxidised sample solutions are again extracted and enriched by SPE, and the LOPs are then separated and detected by ultrahigh-performance liquid chromatography coupled to electrospray ionisation high-resolution mass spectrometry (UHPLC-ESI-HRMS)."

**Referee comment 4:**

At line 6 of page 5, the authors state that samples were stored for several months. Was the conservation of the samples tested somehow?

**Response 4:**

The conservation of the drip water samples was not specifically tested for lignin because the sample collection in the framework of the cave monitoring program took place before the completion of the method development for the analysis of lignin oxidation products. However, lignin is a thermodynamically stable molecule, and the samples were stored with the addition of 5% acetonitrile to prevent any microbial activity, in the dark to prevent photochemical reactions, and at low temperature (4 °C). Therefore, we are convinced that the samples were stable under these conditions.

**Referee comment 5:**

Section 2.2.7, please add the injection volume for analysis and the settings for MS/MS (e.g. collision method and energies).

**Response 5:**

We have added the requested information and we now write in section 2.2.7:

"[...] To separate the LOPs, a Hypersil Gold pentafluorophenyl (PFP) column, 50 mm x 2.1 mm with 1.9  $\mu$ m particle size (also by Thermo Fisher Scientific, Germany) was used. The injection volume was 15  $\mu$ L. A H2O/ACN-gradient program was applied. [...]"

"[...] The mass spectrometer was operated in full scan mode with a resolution of 35 000 and a scan range of m/z 80–500. At the respective retention time windows, the full scan mode was alternated with a targeted MS2-mode with a resolution of 17 500 to identify the LOPs by their specific daughter ions, see Table 1. For the MS2-mode (i.e., parallel reaction monitoring mode in the software XCalibur, provided by Thermo Fisher Scientific), higher-energy collisional dissociation (HCD) was used with 35% normalised collision energy (NCE) for all analytes. The actual collision energy was calculated by the software on the basis of mass and charge of the selected precursor ions and was in the range of 10–14 eV."

Referee comment 6:

I suggest renaming section 3.2.3 "repeatability" as, if I have understood correctly, it describes repetitions with the same equipment and the same operator.

Response 6:

We thank the reviewer for pointing out this mistake. We have renamed the section "repeatability".

Referee comment 7:

Please add the total volume of the surrogate solutions to the caption of figure A2.

Response 7:

We have added the total volume of the surrogate solution and now write in the caption of figure A2:

*"Figure A2.* Recovery rates of the solid phase extraction of LOPs at different spiking concentrations. 20 mL of a surrogate sample solution (2 mol·L-1 NaCl in ultrapure water, acidified with HCl to pH 2) were spiked with 25, 100, 250 and 1000 ng of LOP standards."

Referee comment 8: Typos: Page 3, line 32: H\_2O Page 12, line 17: the symbol sigma should be capitalised.

Response 8: The typos have been corrected.

**Further changes in the manuscript**

There are slight changes in the data of the drip water samples, presented in Table 5 and Figure 8. The reason is that we used an improved integration method to determine the peak area in the chromatograms. However, these changes in the data do not change the interpretation of the data described in section 3.3.3, nor the conclusions of the manuscript.

**"3.3.3 Analysis of cave drip water samples**

Very little is known about how lignin is transported from the soil into the cave and how it is incorporated into a stalagmite. To gain further understanding about these processes, it is useful to also analyze lignin in cave drip water. The lignin concentration in cave drip water is even lower than in stalagmite samples, because crystallization of calcite also serves as an enrichment step for the organic components contained in the water. Therefore, a sample volume of 100–200 mL water was used. Here we show the results of the analysis of six different water samples from the Herbstlabyrinth-Advent-Cave, all sampled in October 2014 (Table 5). As expected, the soil water (SW) has the largest lignin content with 1.8  $\mu$ g·L-1. The rain water (RW) also has a relatively large lignin content of 1.3  $\mu g \cdot L^{-1}$ , which is surprising since this water has not been in contact with soil or vegetation. The lignin content of the cave drip water samples is much lower, ranging from 0.21  $\mu g \cdot L^{-1}$  for the pool water to 0.36  $\mu g \cdot L^{-1}$  for the fast drip site **D1**. The concentrations of all LOPs decrease from the soil water to the cave drip water, but to a different extent. Whereas V-group LOPs and C-group LOPs decrease by 80–92% and 82–90%, respectively, the concentration of S-group LOPs decreases only by 70–76% (Fig. 8). This is also reflected in higher S/V ratios in the cave drip water than in the soil water, with an increasing trend from the soil water over the two fast drip sites D1 and D5 and the slow drip site D2 to the cave pool water. This could be due to different residence times in the cave and the overlaying karst of the water from the different drip sites. These hypotheses should be proven by a further systematic analysis of cave drip water. This would also enable the study of seasonal variations in the lignin input. The monthly cave monitoring program of Mischel et al. (Mischel et al., 2016, 2015) combined with our new method for the analysis of LOPs even in low-concentration cave drip water could be a valuable tool to further investigate these topics."

| Sample                | Sample     | V-group /          | S-group /          | C-group /          | Σ8 /               | C/V         | S/V         |
|-----------------------|------------|--------------------|--------------------|--------------------|--------------------|-------------|-------------|
|                       | volume / L | ng∙L -1 | ng∙L −1 | ng∙L -1 | ng∙L -1 |             |             |
| RW (rain
water)    | 0.185      | 918 ± 69           | 345 ± 31           | 76 ± 17            | 1339 ± 77          | 0.08 ± 0.02 | 0.38 ± 0.04 |
| SW (soil
water)    | 0.076      | 1370 ± 101         | 363 ± 54           | 42 ± 38            | 1775 ± 121         | 0.03 ± 0.03 | 0.26 ± 0.04 |
| D1 (fast
dripping) | 0.265      | 271 ± 21           | 87 ± 16            | 7 ± 11             | 365 ± 29           | 0.03 ± 0.04 | 0.32 ± 0.07 |
| D5 (fast
dripping) | 0.258      | 175 ± 20           | 95 ± 19            | 6 ± 12             | 275 ± 30           | 0.03 ± 0.07 | 0.54 ± 0.13 |
| D2 (slow
dripping) | 0.205      | 157 ± 15           | 107 ± 29           | 4 ± 14             | 269 ± 35           | 0.03 ± 0.09 | 0.68 ± 0.19 |
| PW (pool
water)    | 0.253      | 114 ± 23           | 88 ± 29            | 8 ± 21             | 210 ± 42           | 0.07 ± 0.18 | 0.77 ± 0.29 |

**Table 5**. Concentrations of the V-, S- and C-group LOPs, the sum of all 8 LOPs ( $\Sigma$ 8) and the ratios C/V and S/V in different water samples collected at the Herbstlabyrinth-Advent-Cave in October 2014.

**Figure 8**. LOP concentrations (stacked columns with left axis) and LOP ratios (symbols with right axis) of rain water (RW), soil water (SW), cave drip water from fast drip sites (D1 and D5), a slow drip site (D2) and cave pool water (PW). The stacked columns contain the V-group LOPs (light cyan bars), S-group LOPs (dark cyan bars with vertical stripes) and C-group LOPs (green bars with diagonal stripes). Black triangles show the S/V ratio and blue squares show the C/V ratio.

---

## Author Comment (AC2) · 24 Aug 2018

We thank the two anonymous reviewers for carefully evaluating our manuscript. Our point-by-point reply directly follow the referee comments (blue) and appears in black after each comment. When we refer to text passages of the manuscript we use quotation marks and *cursive font*, new or altered text segments are printed in green.

**Response to Referee #1**

**Referee comment 1:**
1.Elements of scientific novelty should be presented in a more detailed and convincing manner (in the last paragraph of the Introduction).

Response 1:

We have rewritten the last paragraph of the introduction to better point out the scientific novelty of our work. The last paragraph of the introduction now reads:

„*The purpose of this study was to develop and validate a sensitive and selective method for the quantification of LOPs in both speleothem and cave drip water samples using liquid chromatography electrospray ionisation mass spectrometry (LC-ESI-MS). This method offers new possibilities for paleo-vegetation reconstruction since it combines the advantages of lignin analysis as a highly specific vegetation biomarker with the above-mentioned benefits of speleothems as unique terrestrial climate archives. Lignin as a vegetation biomarker is much more specific for higher plants than for example n-alkanes or fatty acids (Jex et al., 2014), and thus can help to interpret other vegetation markers and stable isotope records. Up to now, lignin analysis for paleo vegetation reconstruction has only been applied to lake sediments and peat cores, which contain much larger amounts of organic matter than speleothems. Our method allows to analyse the lignin composition of trace amounts of organic matter preserved in speleothems. The stalagmite samples are first acid digested, and the acidic solution is then extracted by SPE. The eluent is then subjected to CuO oxidation in a microwave assisted digestion method. The oxidised sample solutions are again extracted and enriched by SPE, and the LOPs are then separated and detected by ultrahigh-performance liquid chromatography coupled to electrospray ionisation high-resolution mass spectrometry (UHPLC-ESI-HRMS).*"

**Referee comment 2:**
I suggest that a diagram presenting the steps of the procedure used in the study be added to the EXPERIMENTAL section. It would help understand the details of the analytical protocol better, and allow the written description of the procedure to be shortened.

Response 2:

We thank the reviewer for this helpful suggestion. We added a diagram of the analytical procedure, which is shown below. However, we decided not to shorten the written description of the individual steps of the procedure, because we think that all given details are necessary for the reader to be able to reproduce the analytical method.

[Figure]

**Figure 1.** Process chart of the overall sample preparation procedure. A detailed description of the individual steps is given in section 2.2.

Referee comment 3:
Innovative potential of the results obtained should be explained in detail (CONCLUSIONS).

Response 3:
We have rewritten the conclusion and we now write:

*„Conclusion and outlook*
*We developed a sensitive method for the quantification of LOPs in speleothems and cave drip water and tested it successfully on samples from the Herbstlabyrinth-Advent-Cave. This is, to our knowledge, the first quantitative analysis of LOPs in speleothems and cave drip water. Our method provides a new and highly specific vegetation proxy for the reconstruction of paleo vegetation and paleo climate from speleothem archives. The method was adjusted to the low concentrations of organic matter in speleothems and cave drip water and showed sufficient sensitivity to detect even trace concentrations of lignin. The use of the established CuO oxidation method allows to compare the results to LOP records in other archives. However, as the CuO oxidation step is the main source of variability in our method, an alternative degradation method for lignin with higher reproducibility should be developed. […]"*

Referee comment 4:
Application of proper quality assurance/quality control (QA/QC) procedures is vital for the measurement results to be treated as a source of reliable analytical information. Consequently, I suggest that a separate section devoted to QA/QC be added to the manuscript. Special attention should be paid to: - description of the validation procedure for the applied/proposed analytical protocol, - information on metrological characteristics of the analytical procedure, especially Method Quantitation Limit (MQL) values for the entire procedure (from handling of representative samples to statistical and chemometric evaluation of the data sets obtained), and not only for the analytical techniques used during the analysis of the extracts.

Response 4:
We thank the reviewer for this helpful comment. We have revised section *3.2 Method validation* and expanded it to a QA/QC section. We now write:

*"3.2 Method validation and Quality assurance*

*3.2.1 Selectivity*

*The selectivity of the method was assured by using three parameters for peak identification: the retention time, the exact m/z ratio of the analyte, and the MS²-spectra, as described in section 3.1.1. The variation in the retention time was ± 0.01 min. To assure that the measured peak area was caused only by the analyte, the corresponding peak area of the reagent blank measurement was subtracted.*

*3.2.2 Calibration and linearity*

*External calibration with a standard mixture containing all analytes was performed. The calibration function was obtained using the linear regression method. The parameters of the individual calibration functions are shown in Table A1 in the supplementary information. The concentrations of the standards ranged from 20–500 ng·mL⁻¹ for stalagmite and drip water samples and up to 2000 ng·mL⁻¹ for plant and lignin samples. The calibration was linear in this range.*

*3.2.3 Limits of detection and quantification and reagent blanks*

*The instrumental limits of detection (LOD) and quantification (LOQ) were calculated by using equations (1) and (2), with $\sigma_0$ = standard deviation of the peak area of the solvent blank, or, if no signal was detectable for the solvent blank, of the lowest calibration standard, and the slope of the calibration function, m. The results are shown in Table A1 in the supplementary information.*

*instrumental limit of detection*     $LOD = \dfrac{3.3 \cdot \sigma_0}{m}$ (1)

*instrumental limit of quantification*     $LOQ = \dfrac{10 \cdot \sigma_0}{m}$ (2)

*To eliminate the influence of possible contamination sources on the results, a reagent blank, which had undergone all sample preparations steps, was analyzed with every batch of samples. The concentrations of LOPs measured in this reagent blank were subtracted from the concentrations measured in the samples. The mean values of six reagent blanks measured on different days are shown in Table 2 (the concentrations refer to the final sample solution injected into the LC-MS system). The values ranged from 1.0 ng·mL⁻¹ to 680 ng·mL⁻¹, depending on the analyte (see also 3.2.4). The blank value varied from batch to batch, which is reflected in the standard deviations of the blank values given in Table 2. Therefore, the method detection*

*limit (MDL) and the method quantification limit (MQL) were calculated using only the standard deviation of the peak area of the reagent blank, as shown in equations (3) and (4), with $\sigma_B$ = standard deviation of the peak area of the reagent blank and m = slope of the calibration function. The MDL was below 13.7 ng·mL$^{-1}$ for all relevant analytes and the MQL was below 41.5ng·mL$^{-1}$ for all relevant analytes.*

*method detection limit*     $$MDL = \frac{3.3 \cdot \sigma_B}{m} \qquad\qquad\qquad (3)$$

*method quantification limit*     $$MQL = \frac{10 \cdot \sigma_B}{m} \qquad\qquad\qquad (4)$$

*3.2.4 Origin of blank values*

[revised manuscript text omitted]

*… see Response 5*

Referee comment 5:
I suggest that the protocol described in Journal of Chromatography A (1217, 882-891, 2010) entitled "Estimating uncertainty in analytical procedures based on chromatographic techniques" can be used for evaluation and calculation of expanded uncertainty of results obtained when the procedure described in this manuscript is applied.

Response 5:
We thank the reviewer for this helpful suggestion. We have applied the suggested protocol and have added a subsection on the estimation of uncertainty in section *3.2 Method validation and quality assurance*. We now write:

*"3.2.6    Estimation of uncertainty*

*According to Konieczka and Namiesnik (2010), the main factors contributing to the uncertainty budget are the uncertainty of the measurement of the weight or volume of the sample, $u_r(sample)$, the repeatability of the sample preparation procedure, $u_r(rep.)$, the recovery determination of the internal standard, $u_r(recov.)$, the calibration step, $u_r(cal.)$, and the uncertainty associated with analyte concentrations close to the limit of detection, $u_r(LOD)$. The combined relative uncertainty $U_r$ is expressed in equation 5.*

$$U_r = \sqrt{\left(u_r(sample)\right)^2 + \left(u_r(rep.)\right)^2 + \left(u_r(recov.)\right)^2 + \left(u_r(cal.)\right)^2 + \left(u_r(LOD)\right)^2} \qquad (5)$$

*In our method, $u_r(sample)$ is relatively small with 1 mg or 1 mL, which is usually < 1%. The uncertainty associated with the repeatability of the sample preparation, calculated as the standard deviation of three individually prepared subsamples as explained in section 3.2.5, has the largest influence and can equal 1–30%. The uncertainty of the recovery determination of the internal standard, calculated as the standard deviation of the internal standard, contributes with 1–6%. $u_r(cal.)$, calculated as the standard deviation of the concentration determination of three injections of the same sample into the LC-MS-system, can equal 1–15%, but is usually around 3–5%. $u_r(LOD)$, calculated according to equation (6), depends strongly on the concentration c of the analyte.*

$$u_r = \frac{LOD}{c} \qquad (6)$$

*In the data for stalagmite samples presented in Table 2, $u_r$(LOD) equals 0.1–5% for most analytes, 17% for Sal and 27–100% for the p-hydroxy group.*
*The errors for all results presented in this work were calculated using the law of propagation of uncertainty. All equations used for calculating concentrations, lignin oxidation parameters and errors are shown in section A4 in the supplementary information."*

Referee comment 6:

Green aspects of different approaches known from the literature should be discussed. There is a strong need of insertation of an additional chapter to the text of the paper. In this paper the newest literature information on the development of green analytical principles and approaches should be presented. Green analytical Chemistry (GAC) should be treated as a very important part of green chemistry. Authors should study the literature data this field in deeper manner.

Response 6:

We thank the reviewer for this interesting and helpful comment. We have studied the recent literature on Green analytical Chemistry and added an additional section about this topic. We now write:

*"Aspects of green analytical chemistry*
*When developing a new analytical method, it is advantageous to consider how environment-friendly (green) the different approaches are. The principles of green analytical chemistry include, among others, to generate as little waste as possible, to eliminate or replace toxic reagents, to miniaturize analytical instruments or to avoid derivatization (Gałuszka et al., 2013; Armenta et al., 2008). In our method, we tried to favour greener approaches over less green approaches whenever possible without sacrificing other qualities like sensitivity. We used solid phase extraction, which consumes considerably less solvent than liquid-liquid extraction, and UHPLC, which is less solvent and time consuming than HPLC. In addition, liquid chromatography does not require a derivatization step, as opposed to gas chromatography. However, the least green step in our method is the CuO oxidation step, as it generates toxic waste and consumes energy. We still chose the CuO oxidation method for our proof of principle analysis because it is the most widely used lignin degradation method for the analysis of LOPs and therefore allows us to compare our results with existing LOP records. In the future, however, a greener approach to the degradation of lignin to LOPs should be chosen, which could, for example, be based on electrolysis, preferably in a miniaturized flow cell (Leppla, 2016)."*

**Response to Referee #2**

Referee comment 1:
I agree with reviewer #1 concerning the suggestion to add a QA/QC section and a diagram/scheme of the sample preparation protocol.

Response 1:
As already mentioned above, we have added a QA/QC section, which is described in detail in the response to Referee #1, as well as a diagram/scheme of the sample preparation protocol.

Referee comment 2:
I suggest using consistent unit measures when reporting concentration values throughout the manuscript, especially concerning Table 2, Table 3 and section 3.3.1. I would also report LODs and LOQs in terms of concentrations rather than absolute amounts as it may be ambiguous if the absolute amount is referred to the total amount of samples or the total amount of analyte injected in the instrument.

Response 2:
We thank the reviewer for pointing out this inconsistency in unit measures. We now report all concentrations referring to solid samples (stalagmite, plant samples, lignin powder) in ng/g, µg/g or mg/g, all concentrations referring to drip water samples in ng/L, and all concentrations referring to the final sample solution injected into the instrument (standards, blanks, LODs, LOQs) in ng/mL. In case of ambiguity, we specify what the concentration refers to. We do not use absolute amounts anymore.

Referee comment 3:
I would suggest removing the first paragraph of the introduction or, alternatively, combined it with the last paragraph of the introduction.

Response 3:
We have removed the first paragraph of the introduction and have combined it with the last paragraph of the introduction. The introduction now starts directly with the sentence:

*"Speleothems are calcareous mineral deposits that form within caves in karstified carbonate rock. The most common types of speleothems are…".*

The last paragraph of the introduction now reads:

*„The purpose of this study was to develop and validate a sensitive and selective method for the quantification of LOPs in both speleothem and cave drip water samples using liquid chromatography electrospray ionisation mass spectrometry (LC-ESI-MS). This method offers new possibilities for paleo vegetation reconstruction since it combines the advantages of lignin analysis as a highly specific vegetation biomarker with the above-mentioned benefits of speleothems as unique terrestrial climate archives. Lignin as a vegetation biomarker is much more specific for higher plants than for example n-alkanes or fatty acids (Jex et al., 2014), and thus can help to interpret other vegetation markers and stable isotope records. Up to now, lignin*

*analysis for paleo vegetation reconstruction was only applied to lake sediments and peat cores, which contain much larger amounts of organic matter than speleothems. Our method allows to analyse the lignin composition of trace amounts of organic matter preserved in speleo-thems. The stalagmite samples are first acid digested, and the acidic solution is then extracted by SPE. The eluent is then subjected to CuO oxidation in a microwave assisted digestion method. The oxidised sample solutions are again extracted and enriched by SPE, and the LOPs are then separated and detected by ultrahigh-performance liquid chromatography coupled to electrospray ionisation high-resolution mass spectrometry (UHPLC-ESI-HRMS)."*

**Referee comment 4:**
At line 6 of page 5, the authors state that samples were stored for several months. Was the conservation of the samples tested somehow?

Response 4:
The conservation of the drip water samples was not specifically tested for lignin because the sample collection in the framework of the cave monitoring program took place before the completion of the method development for the analysis of lignin oxidation products. However, lignin is a thermodynamically stable molecule, and the samples were stored with the addition of 5% acetonitrile to prevent any microbial activity, in the dark to prevent photochemical reactions, and at low temperature (4 °C). Therefore, we are convinced that the samples were stable under these conditions.

**Referee comment 5:**
Section 2.2.7, please add the injection volume for analysis and the settings for MS/MS (e.g. collision method and energies).

Response 5:
We have added the requested information and we now write in section 2.2.7:

*"[…] To separate the LOPs, a Hypersil Gold pentafluorophenyl (PFP) column, 50 mm x 2.1 mm with 1.9 µm particle size (also by Thermo Fisher Scientific, Germany) was used. The injection volume was 15 µL. A H2O/ACN-gradient program was applied. […]"*

*"[…] The mass spectrometer was operated in full scan mode with a resolution of 35 000 and a scan range of m/z 80–500. At the respective retention time windows, the full scan mode was alternated with a targeted $MS^2$-mode with a resolution of 17 500 to identify the LOPs by their specific daughter ions, see Table 1. For the $MS^2$-mode (i.e., parallel reaction monitoring mode in the software XCalibur, provided by Thermo Fisher Scientific), higher-energy collisional disso-ciation (HCD) was used with 35% normalised collision energy (NCE) for all analytes. The actual collision energy was calculated by the software on the basis of mass and charge of the selected precursor ions and was in the range of 10–14 eV."*

I suggest renaming section 3.2.3 "repeatability" as, if I have understood correctly, it describes repetitions with the same equipment and the same operator.

Response 6:
We thank the reviewer for pointing out this mistake. We have renamed the section "repeatability".

Referee comment 7:
Please add the total volume of the surrogate solutions to the caption of figure A2.

Response 7:
We have added the total volume of the surrogate solution and now write in the caption of figure A2:

„**Figure A2.** *Recovery rates of the solid phase extraction of LOPs at different spiking concentrations. 20 mL of a surrogate sample solution (2 mol·L$^{-1}$ NaCl in ultrapure water, acidified with HCl to pH 2) were spiked with 25, 100, 250 and 1000 ng of LOP standards.*"

Referee comment 8:
Typos: Page 3, line 32: H_2O Page 12, line 17: the symbol sigma should be capitalised.

Response 8:
The typos have been corrected.

**Further changes in the manuscript**

There are slight changes in the data of the drip water samples, presented in Table 5 and Figure 8. The reason is that we used an improved integration method to determine the peak area in the chromatograms. However, these changes in the data do not change the interpretation of the data described in section 3.3.3, nor the conclusions of the manuscript.

*"3.3.3 Analysis of cave drip water samples*

*Very little is known about how lignin is transported from the soil into the cave and how it is incorporated into a stalagmite. To gain further understanding about these processes, it is useful to also analyze lignin in cave drip water. The lignin concentration in cave drip water is even lower than in stalagmite samples, because crystallization of calcite also serves as an enrichment step for the organic components contained in the water. Therefore, a sample volume of 100–200 mL water was used. Here we show the results of the analysis of six different water samples from the Herbstlabyrinth-Advent-Cave, all sampled in October 2014 (Table 5). As expected, the soil water (SW) has the largest lignin content with $1.8\ \mu g \cdot L^{-1}$. The rain water (RW) also has a relatively large lignin content of $1.3\ \mu g \cdot L^{-1}$, which is surprising since this water has not been in contact with soil or vegetation. The lignin content of the cave drip water samples is much lower, ranging from $0.21\ \mu g \cdot L^{-1}$ for the pool water to $0.36\ \mu g \cdot L^{-1}$ for the fast drip site D1. The concentrations of all LOPs decrease from the soil water to the cave drip water, but to a different extent. Whereas V-group LOPs and C-group LOPs decrease by 80–92% and 82–90%, respectively, the concentration of S-group LOPs decreases only by 70–76% (Fig. 8). This is also reflected in higher S/V ratios in the cave drip water than in the soil water, with an increasing trend from the soil water over the two fast drip sites D1 and D5 and the slow drip site D2 to the cave pool water. This could be due to different residence times in the cave and the overlaying karst of the water from the different drip sites. These hypotheses should be proven by a further systematic analysis of cave drip water. This would also enable the study of seasonal variations in the lignin input. The monthly cave monitoring program of Mischel et al. (Mischel et al., 2016, 2015) combined with our new method for the analysis of LOPs even in low-concentration cave drip water could be a valuable tool to further investigate these topics."*

**Table 5**. Concentrations of the V-, S- and C-group LOPs, the sum of all 8 LOPs (Σ8) and the ratios C/V and S/V in different water samples collected at the Herbstlabyrinth-Advent-Cave in October 2014.

| Sample | Sample volume / L | V-group / ng·L$^{-1}$ | S-group / ng·L$^{-1}$ | C-group / ng·L$^{-1}$ | Σ8 / ng·L$^{-1}$ | C/V | S/V |
|---|---|---|---|---|---|---|---|
| RW (rain water) | 0.185 | 918 ± 69 | 345 ± 31 | 76 ± 17 | 1339 ± 77 | 0.08 ± 0.02 | 0.38 ± 0.04 |
| SW (soil water) | 0.076 | 1370 ± 101 | 363 ± 54 | 42 ± 38 | 1775 ± 121 | 0.03 ± 0.03 | 0.26 ± 0.04 |
| D1 (fast dripping) | 0.265 | 271 ± 21 | 87 ± 16 | 7 ± 11 | 365 ± 29 | 0.03 ± 0.04 | 0.32 ± 0.07 |
| D5 (fast dripping) | 0.258 | 175 ± 20 | 95 ± 19 | 6 ± 12 | 275 ± 30 | 0.03 ± 0.07 | 0.54 ± 0.13 |
| D2 (slow dripping) | 0.205 | 157 ± 15 | 107 ± 29 | 4 ± 14 | 269 ± 35 | 0.03 ± 0.09 | 0.68 ± 0.19 |
| PW (pool water) | 0.253 | 114 ± 23 | 88 ± 29 | 8 ± 21 | 210 ± 42 | 0.07 ± 0.18 | 0.77 ± 0.29 |

[Figure]

***Figure 8**. LOP concentrations (stacked columns with left axis) and LOP ratios (symbols with right axis) of rain water (RW), soil water (SW), cave drip water from fast drip sites (D1 and D5), a slow drip site (D2) and cave pool water (PW). The stacked columns contain the V-group LOPs (light cyan bars), S-group LOPs (dark cyan bars with vertical stripes) and C-group LOPs (green bars with diagonal stripes). Black triangles show the S/V ratio and blue squares show the C/V ratio.*

---

## Author Response (AR1)

**Author's response**

We thank the two anonymous reviewers for carefully evaluating our manuscript. Our point-by-point reply directly follow the referee comments (blue) and appears in black after each comment. When we refer to text passages of the manuscript we use quotation marks and *cursive font*, new or altered text segments are printed in green.

**Response to Referee #1**

Referee comment 1:

1.Elements of scientific novelty should be presented in a more detailed and convincing manner (in the last paragraph of the Introduction).

Response 1:

We have rewritten the last paragraph of the introduction to better point out the scientific novelty of our work. The last paragraph of the introduction now reads:

*„The purpose of this study was to develop and validate a sensitive and selective method for the quantification of LOPs in both speleothem and cave drip water samples using liquid chromatography electrospray ionisation mass spectrometry (LC-ESI-MS). This method offers new possibilities for paleo-vegetation reconstruction since it combines the advantages of lignin analysis as a highly specific vegetation biomarker with the above-mentioned benefits of speleothems as unique terrestrial climate archives. Lignin as a vegetation biomarker is much more specific for higher plants than for example n-alkanes or fatty acids (Jex et al., 2014), and thus can help to interpret other vegetation markers and stable isotope records. Up to now, lignin analysis for paleo vegetation reconstruction has only been applied to lake sediments and peat cores, which contain much larger amounts of organic matter than speleothems. Our method allows to analyse the lignin composition of trace amounts of organic matter preserved in speleothems. The stalagmite samples are first acid digested, and the acidic solution is then extracted by SPE. The eluent is then subjected to CuO oxidation in a microwave assisted digestion method. The oxidised sample solutions are again extracted and enriched by SPE, and the LOPs are then separated and detected by ultrahigh-performance liquid chromatography coupled to electrospray ionisation high-resolution mass spectrometry (UHPLC-ESI-HRMS)."*

Referee comment 2:

I suggest that a diagram presenting the steps of the procedure used in the study be added to the EXPERIMENTAL section. It would help understand the details of the analytical protocol better, and allow the written description of the procedure to be shortened.

Response 2:

We thank the reviewer for this helpful suggestion. We added a diagram of the analytical procedure, which is shown below. However, we decided not to shorten the written description of the individual steps of the procedure, because we think that all given details are necessary for the reader to be able to reproduce the analytical method.

[Figure]

**Figure 1.** Process chart of the overall sample preparation procedure. A detailed description of the individual steps is given in section 2.2.

Referee comment 3:
Innovative potential of the results obtained should be explained in detail (CONCLUSIONS).

Response 3:
We have rewritten the conclusion and we now write:

*„Conclusion and outlook*
*We developed a sensitive method for the quantification of LOPs in speleothems and cave drip water and tested it successfully on samples from the Herbstlabyrinth-Advent-Cave. This is, to our knowledge, the first quantitative analysis of LOPs in speleothems and cave drip water. Our method provides a new and highly specific vegetation proxy for the reconstruction of paleo vegetation and paleo climate from speleothem archives. The method was adjusted to the low concentrations of organic matter in speleothems and cave drip water and showed sufficient sensitivity to detect even trace concentrations of lignin. The use of the established CuO oxidation method allows to compare the results to LOP records in other archives. However, as the CuO oxidation step is the main source of variability in our method, an alternative degradation method for lignin with higher reproducibility should be developed. […]"*

Referee comment 4:
Application of proper quality assurance/quality control (QA/QC) procedures is vital for the measurement results to be treated as a source of reliable analytical information. Consequently, I suggest that a separate section devoted to QA/QC be added to the manuscript. Special attention should be paid to: - description of the validation procedure for the applied/proposed analytical protocol, - information on metrological characteristics of the analytical procedure, especially Method Quantitation Limit (MQL) values for the entire procedure (from handling of representative samples to statistical and chemometric evaluation of the data sets obtained), and not only for the analytical techniques used during the analysis of the extracts.

Response 4:
We thank the reviewer for this helpful comment. We have revised section *3.2 Method validation* and expanded it to a QA/QC section. We now write:

*"3.2 Method validation and Quality assurance*

*3.2.1 Selectivity*

*The selectivity of the method was assured by using three parameters for peak identification: the retention time, the exact m/z ratio of the analyte, and the $MS^2$-spectra, as described in section 3.1.1. The variation in the retention time was ± 0.01 min. To assure that the measured peak area was caused only by the analyte, the corresponding peak area of the reagent blank measurement was subtracted.*

*3.2.2 Calibration and linearity*

*External calibration with a standard mixture containing all analytes was performed. The calibration function was obtained using the linear regression method. The parameters of the individual calibration functions are shown in Table A1 in the supplementary information. The concentrations of the standards ranged from 20–500 ng·mL$^{-1}$ for stalagmite and drip water samples and up to 2000 ng·mL$^{-1}$ for plant and lignin samples. The calibration was linear in this range.*

*3.2.3 Limits of detection and quantification and reagent blanks*

*The instrumental limits of detection (LOD) and quantification (LOQ) were calculated by using equations (1) and (2), with $\sigma_0$ = standard deviation of the peak area of the solvent blank, or, if no signal was detectable for the solvent blank, of the lowest calibration standard, and the slope of the calibration function, m. The results are shown in Table A1 in the supplementary information.*

*instrumental limit of detection* $\quad LOD = \frac{3.3 \cdot \sigma_0}{m}$ (1)

*instrumental limit of quantification* $\quad LOQ = \frac{10 \cdot \sigma_0}{m}$ (2)

*To eliminate the influence of possible contamination sources on the results, a reagent blank, which had undergone all sample preparations steps, was analyzed with every batch of samples. The concentrations of LOPs measured in this reagent blank were subtracted from the concentrations measured in the samples. The mean values of six reagent blanks measured on different days are shown in Table 2 (the concentrations refer to the final sample solution injected into the LC-MS system). The values ranged from 1.0 ng·mL$^{-1}$ to 680 ng·mL$^{-1}$, depending on the analyte (see also 3.2.4). The blank value varied from batch to batch, which is reflected in the standard deviations of the blank values given in Table 2. Therefore, the method detection*

*limit (MDL) and the method quantification limit (MQL) were calculated using only the standard deviation of the peak area of the reagent blank, as shown in equations (3) and (4), with $\sigma_B$ = standard deviation of the peak area of the reagent blank and m = slope of the calibration function. The MDL was below 13.7 ng·mL$^{-1}$ for all relevant analytes and the MQL was below 41.5ng·mL$^{-1}$ for all relevant analytes.*

*method detection limit* $\qquad MDL = \frac{3.3 \cdot \sigma_B}{m}$ $\hspace{4cm}$ *(3)*

*method quantification limit* $\qquad MQL = \frac{10 \cdot \sigma_B}{m}$ $\hspace{3.5cm}$ *(4)*

*3.2.4 Origin of blank values*

[revised manuscript text omitted]

*3.2.6    Estimation of uncertainty"*

*… see Response 5*

Referee comment 5:
I suggest that the protocol described in Journal of Chromatography A (1217, 882-891, 2010) entitled "Estimating uncertainty in analytical procedures based on chromatographic techniques" can be used for evaluation and calculation of expanded uncertainty of results obtained when the procedure described in this manuscript is applied.

Response 5:
We thank the reviewer for this helpful suggestion. We have applied the suggested protocol and have added a subsection on the estimation of uncertainty in section *3.2 Method validation and quality assurance*. We now write:

*"3.2.6    Estimation of uncertainty*

*According to Konieczka and Namiesnik (2010), the main factors contributing to the uncertainty budget are the uncertainty of the measurement of the weight or volume of the sample, $u_r$(sample), the repeatability of the sample preparation procedure, $u_r$(rep.), the recovery determination of the internal standard, $u_r$(recov.), the calibration step, $u_r$(cal.), and the uncertainty associated with analyte concentrations close to the limit of detection, $u_r$(LOD). The combined relative uncertainty $U_r$ is expressed in equation 5.*

$$U_r = \sqrt{\left(u_r(sample)\right)^2 + \left(u_r(rep.)\right)^2 + \left(u_r(recov.)\right)^2 + \left(u_r(cal.)\right)^2 + \left(u_r(LOD)\right)^2} \qquad (5)$$

*In our method, $u_r$(sample) is relatively small with 1 mg or 1 mL, which is usually < 1%. The uncertainty associated with the repeatability of the sample preparation, calculated as the standard deviation of three individually prepared subsamples as explained in section 3.2.5, has the largest influence and can equal 1–30%. The uncertainty of the recovery determination of the internal standard, calculated as the standard deviation of the internal standard, contributes with 1–6%. $u_r$(cal.), calculated as the standard deviation of the concentration determination of three injections of the same sample into the LC-MS-system, can equal 1–15%, but is usually around 3–5%. $u_r$(LOD), calculated according to equation (6), depends strongly on the concentration c of the analyte.*

$$u_r = \frac{LOD}{c} \qquad (6)$$

*In the data for stalagmite samples presented in Table 2, $u_r$(LOD) equals 0.1–5% for most analytes, 17% for Sal and 27–100% for the p-hydroxy group.*
*The errors for all results presented in this work were calculated using the law of propagation of uncertainty. All equations used for calculating concentrations, lignin oxidation parameters and errors are shown in section A4 in the supplementary information."*

Referee comment 6:

Green aspects of different approaches known from the literature should be discussed. There is a strong need of insertation of an additional chapter to the text of the paper. In this paper the newest literature information on the development of green analytical principles and approaches should be presented. Green analytical Chemistry (GAC) should be treated as a very important part of green chemistry. Authors should study the literature data this field in deeper manner.

Response 6:

We thank the reviewer for this interesting and helpful comment. We have studied the recent literature on Green analytical Chemistry and added an additional section about this topic. We now write:

*"Aspects of green analytical chemistry*
*When developing a new analytical method, it is advantageous to consider how environment-friendly (*green*) the different approaches are. The principles of green analytical chemistry include, among others, to generate as little waste as possible, to eliminate or replace toxic reagents, to miniaturize analytical instruments or to avoid derivatization (Gałuszka et al., 2013; Armenta et al., 2008). In our method, we tried to favour greener approaches over less green approaches whenever possible without sacrificing other qualities like sensitivity. We used solid phase extraction, which consumes considerably less solvent than liquid-liquid extraction, and UHPLC, which is less solvent and time consuming than HPLC. In addition, liquid chromatography does not require a derivatization step, as opposed to gas chromatography. However, the least green step in our method is the CuO oxidation step, as it generates toxic waste and consumes energy. We still chose the CuO oxidation method for our proof of principle analysis because it is the most widely used lignin degradation method for the analysis of LOPs and therefore allows us to compare our results with existing LOP records. In the future, however, a greener approach to the degradation of lignin to LOPs should be chosen, which could, for example, be based on electrolysis, preferably in a miniaturized flow cell (Leppla, 2016)."*

**Response to Referee #2**

Referee comment 1:
I agree with reviewer #1 concerning the suggestion to add a QA/QC section and a dia-gram/scheme of the sample preparation protocol.

Response 1:
As already mentioned above, we have added a QA/QC section, which is described in detail in the response to Referee #1, as well as a diagram/scheme of the sample preparation protocol.

Referee comment 2:
I suggest using consistent unit measures when reporting concentration values throughout the manuscript, especially concerning Table 2, Table 3 and section 3.3.1. I would also report LODs and LOQs in terms of concentrations rather than absolute amounts as it may be ambiguous if the absolute amount is referred to the total amount of samples or the total amount of analyte injected in the instrument.

Response 2:
We thank the reviewer for pointing out this inconsistency in unit measures. We now report all concentrations referring to solid samples (stalagmite, plant samples, lignin powder) in ng/g, µg/g or mg/g, all concentrations referring to drip water samples in ng/L, and all concentrations referring to the final sample solution injected into the instrument (standards, blanks, LODs, LOQs) in ng/mL. In case of ambiguity, we specify what the concentration refers to. We do not use absolute amounts anymore.

Referee comment 3:
I would suggest removing the first paragraph of the introduction or, alternatively, combined it with the last paragraph of the introduction.

Response 3:
We have removed the first paragraph of the introduction and have combined it with the last paragraph of the introduction. The introduction now starts directly with the sentence:

*"Speleothems are calcareous mineral deposits that form within caves in karstified carbonate rock. The most common types of speleothems are…".*

The last paragraph of the introduction now reads:

*"The purpose of this study was to develop and validate a sensitive and selective method for the quantification of LOPs in both speleothem and cave drip water samples using liquid chro-matography electrospray ionisation mass spectrometry (LC-ESI-MS). This method offers new possibilities for paleo vegetation reconstruction since it combines the advantages of lignin analysis as a highly specific vegetation biomarker with the above-mentioned benefits of spele-othems as unique terrestrial climate archives. Lignin as a vegetation biomarker is much more specific for higher plants than for example n-alkanes or fatty acids (Jex et al., 2014), and thus can help to interpret other vegetation markers and stable isotope records. Up to now, lignin*

*analysis for paleo vegetation reconstruction was only applied to lake sediments and peat cores, which contain much larger amounts of organic matter than speleothems. Our method allows to analyse the lignin composition of trace amounts of organic matter preserved in speleothems.* The stalagmite samples are first acid digested, and the acidic solution is then extracted by SPE. The eluent is then subjected to CuO oxidation in a microwave assisted digestion method. The oxidised sample solutions are again extracted and enriched by SPE, and the LOPs are then separated and detected by ultrahigh-performance liquid chromatography coupled to electrospray ionisation high-resolution mass spectrometry (UHPLC-ESI-HRMS)."

**Referee comment 4:**
At line 6 of page 5, the authors state that samples were stored for several months. Was the conservation of the samples tested somehow?

Response 4:
The conservation of the drip water samples was not specifically tested for lignin because the sample collection in the framework of the cave monitoring program took place before the completion of the method development for the analysis of lignin oxidation products. However, lignin is a thermodynamically stable molecule, and the samples were stored with the addition of 5% acetonitrile to prevent any microbial activity, in the dark to prevent photochemical reactions, and at low temperature (4 °C). Therefore, we are convinced that the samples were stable under these conditions.

**Referee comment 5:**
Section 2.2.7, please add the injection volume for analysis and the settings for MS/MS (e.g. collision method and energies).

Response 5:
We have added the requested information and we now write in section 2.2.7:

"[…] To separate the LOPs, a Hypersil Gold pentafluorophenyl (PFP) column, 50 mm x 2.1 mm with 1.9 µm particle size (also by Thermo Fisher Scientific, Germany) was used. *The injection volume was 15 µL.* A H2O/ACN-gradient program was applied. […]"

"[…] The mass spectrometer was operated in full scan mode with a resolution of *35 000* and a scan range of m/z 80–500. At the respective retention time windows, the full scan mode was alternated with a targeted $MS^2$-mode with a resolution of 17 500 to identify the LOPs by their specific daughter ions, see Table 1. *For the $MS^2$-mode (i.e., parallel reaction monitoring mode in the software XCalibur, provided by Thermo Fisher Scientific), higher-energy collisional dissociation (HCD) was used with 35% normalised collision energy (NCE) for all analytes. The actual collision energy was calculated by the software on the basis of mass and charge of the selected precursor ions and was in the range of 10–14 eV.*"

Referee comment 6:
I suggest renaming section 3.2.3 "repeatability" as, if I have understood correctly, it describes repetitions with the same equipment and the same operator.

Response 6:
We thank the reviewer for pointing out this mistake. We have renamed the section "repeatability".

Referee comment 7:
Please add the total volume of the surrogate solutions to the caption of figure A2.

Response 7:
We have added the total volume of the surrogate solution and now write in the caption of figure A2:

„*Figure A2. Recovery rates of the solid phase extraction of LOPs at different spiking concentrations. 20 mL of a surrogate sample solution (2 mol·L$^{-1}$ NaCl in ultrapure water, acidified with HCl to pH 2) were spiked with 25, 100, 250 and 1000 ng of LOP standards.*"

Referee comment 8:
Typos: Page 3, line 32: H_2O Page 12, line 17: the symbol sigma should be capitalised.

Response 8:
The typos have been corrected.

**Further changes in the manuscript**

There are slight changes in the data of the drip water samples, presented in Table 5 and Figure 8. The reason is that we used an improved integration method to determine the peak area in the chromatograms. However, these changes in the data do not change the interpretation of the data described in section 3.3.3, nor the conclusions of the manuscript.

[revised manuscript text omitted]

$$\text{instrumental limit of detection} \qquad \text{LOD} = \frac{3.3 \cdot \sigma_0}{m} \tag{1}$$

5 $$\text{instrumental limit of quantification} \qquad \text{LOQ} = \frac{10 \cdot \sigma_0}{m} \tag{2}$$

To eliminate the influence of possible contamination sources on the results, a  reagent blank, which had undergone all sample preparations steps, was  analyzed with every batch of samples. The concentrations of LOPs measured in this reagent blank were subtracted from the concentrations measured in the samples. The mean values of six  reagent blanks measured on different days are shown in Table 2 (the concentrations refer to the final sample solution injected into

10 the LC-MS system). The values ranged from 1.0 ng $\cdot$ mL$^{-1}$ to 680 ng $\cdot$ mL$^{-1}$, depending on the analyte (see also 3.2.4). The blank value varied from batch to batch, which is reflected in the standard deviations of the blank values given in Table 2. Therefore, the method detection limit (MDL) and the method quantification limit (MQL) were calculated  using only the standard deviation of the peak area of the reagent blank, as shown in equations (3) and (4),  with $\sigma_B$

15 $=$ standard deviation of the peak area of the reagent blank and $m$ = slope of the calibration function. The MDL was below 13.7 ng $\cdot$ mL$^{-1}$ for all relevant analytes and the  MQL was below 41.5 ng $\cdot$ mL$^{-1}$ for all relevant analytes.

$$\text{\sout{LOD}method detection limit} \qquad \text{MDL} = \frac{3 \cdot \sigma - b}{m}\frac{3.3 \cdot \sigma_B}{m} \qquad \text{with } \sigma = \text{standard deviation of the blank value, } b = \text{intersect with the} \tag{3}$$

20 $$\text{\sout{LOQ}method quantification limit} \qquad \text{MQL} = \frac{10 \cdot \sigma - b}{m}\frac{10 \cdot \sigma_B}{m} \tag{4}$$

**3.2.4  Origin of blank values**

The blank values shown in Table  2 reflect the natural occurrence of the different analytes. The highest blank values have been found for the p-hydroxy group, p-coumaric acid, cinnamic acid, vanillin and vanillic acid. The p-hydroxy group is known to originate not only from lignin, but also from protein rich material such as bacteria (Jex et al., 2014). For p-hydroxy acetophe-

25 none, which has a lower blank value than p-hydroxy benzoic acid and p-hydroxy benzaldehyde, it is in discussion whether it

Table 2. Method detection  after subtraction of the reagent blank (MDL) in ng·mL⁻¹, method quantification  after subtraction of the reagent blank (MQL) in ng·mL⁻¹, mean value of three subsamples of 3.4 g stalagmite after blank subtraction in ng·mL⁻¹ and in ng·g⁻¹ of the initial stalagmite sample, mean blank value of six reagent blanks measured on different days in ng·mL⁻¹, and recovery values of the SPE procedure to extract LOPs (Recov. SPE) in %. All concentrations in ng·mL⁻¹ refer to the final sample solution injected into the LC-MS system. The errors stated in this table are standard deviations of $n$ samples. For the methods of calculation used please refer to the text. The abbreviations for the analytes are shown in Table 1.

| analyte | MDL / ng·mL⁻¹ | MQL / ng·mL⁻¹ | Mean stalagmite / ng·mL⁻¹ (n=3) | Mean stalagmite / ng·g⁻¹ (n=3) | Mean blank / ng·
[revised manuscript text omitted]

**Table A1.** Linear regresssion parameters of the external calibration functions and instrumental limits of detection (LOD) and qualibration (LOQ).

| analyte | $R^2$ | slope | intersept | instrumental LOD / $\text{ng} \cdot \text{mL}^{-1}$ | instrumental LOQ / $\text{ng} \cdot \text{mL}^{-1}$ |
|---------|-------|-------|-----------|------|------|
| pHac | 0.9998 | 6371269 | 229351 | 0.39 | 1.19 |
| pHal | 0.9949 | 24875695 | 10116687 | 0.05 | 0.15 |
| pHon | 0.9988 | 14313979 | 1679485 | 0.18 | 0.55 |
| Vac | 0.9998 | 206028 | 3287 | 0.48 | 1.46 |
| Val | 0.9997 | 630114 | 39639 | 0.25 | 0.75 |
| Von | 0.9962 | 129627 | -6652 | 2.27 | 6.89 |
| Sac | 0.9993 | 270474 | -23398 | 0.55 | 1.66 |
| Sal | 0.9996 | 170558 | -17658 | 3.75 | 11.36 |
| Son | 0.9998 | 122729 | 3837 | 7.92 | 24.00 |
| pCac | 0.9978 | 11971947 | 2189589 | 1.08 | 3.26 |
| Fac | 0.9998 | 2028750 | -169451 | 0.10 | 0.29 |
| Eval | 0.9998 | 1769186 | 929 | 0.24 | 0.73 |
| Ciac | 0.9996 | 100371 | 7569 | 4.44 | 13.45 |

**A4 Equations used for calculation of concentrations, lignin oxidation parameters and errors bars**

The concentration $c(\text{analyte})$ of real samples was calculated by equation (A1), with $A =$ mean peak area of three LC-MS analyses of the sample, $B =$ mean peak area of three LC-MS analyses of the blank sample, $b =$ intersect of the Y-axis of the external calibration curve, $m =$ slope of the external calibration curve, $f_r =$ recovery factor of the internal standard ethylvanillin (Eval), see equation (A2), $V =$ volume of the final sample solution and $m_{\text{sample}} =$ sample mass.

$$c(\text{analyte}) = \frac{A - B - b}{m} \cdot \frac{1}{f_r} \cdot \frac{V}{m_{\text{sample}}} \tag{A1}$$

$$f_r = \frac{c(\text{Eval})_{\text{measured}}}{c(\text{Eval})_{\text{spiked}}} \tag{A2}$$

The error $\Delta c(\text{analyte})$ of the concentration $c(\text{analyte})$ was calculated by equation (A3).

$$\Delta c(\text{analyte}) = \sqrt{\left(\frac{\partial A}{\partial c}\Delta A\right)^2 + \left(\frac{\partial B}{\partial c}\Delta B\right)^2 + \left(\frac{\partial b}{\partial c}\Delta b\right)^2 + \left(\frac{\partial m}{\partial c}\Delta m\right)^2 + \left(\frac{\partial f_r}{\partial c}\Delta f_r\right)^2 + \left(\frac{\partial m_{\text{sample}}}{\partial c}\Delta m_{\text{sample}}\right)^2} \tag{A3}$$

The lignin oxidation parameters were calculated according to equations (A4) to (A9). Their errors were calculated using the law of propagation of uncertainty (equations not shown).

$$\text{C-group LOPs} = c(\text{p-Cac}) + c(\text{t-Fac}) \tag{A4}$$

$$\text{S-group LOPs} = c(\text{Sac}) + c(\text{Sal}) + c(\text{Son}) \tag{A5}$$

5  $$\text{V-group LOPs} = c(\text{Vac}) + c(\text{Val}) + c(\text{Von}) \tag{A6}$$

$$\Sigma 8 = \text{C-group LOPs} + \text{S-group LOPs} + \text{V-group LOPs} \tag{A7}$$

$$\text{C/V} = \frac{\text{C-group LOPs}}{\text{V-group LOPs}} \tag{A8}$$

$$\text{S/V} = \frac{\text{S-group LOPs}}{\text{V-group LOPs}} \tag{A9}$$

[revised manuscript text omitted]